# Intracellular bound chlorophyll residues identify 1 Gyr-old fossils as eukaryotic algae

Marie Catherine Sforna[1✉], Corentin C. Loron [1], Catherine F. Demoulin[1], Camille François[1,2], Yohan Cornet[1], Yannick J. Lara[1], Daniel Grolimund [3], Dario Ferreira Sanchez [3], Kadda Medjoubi[4], Andrea Somogyi [4], Ahmed Addad[5], Alexandre Fadel[5], Philippe Compère[6], Daniel Baudet [7], Jochen J. Brocks [8] & Emmanuelle J. Javaux [1✉]

The acquisition of photosynthesis is a fundamental step in the evolution of eukaryotes. However, few phototrophic organisms are unambiguously recognized in the Precambrian record. The in situ detection of metabolic byproducts in individual microfossils is the key for the direct identification of their metabolisms. Here, we report a new integrative methodology using synchrotron-based X-ray fluorescence and absorption. We evidence bound nickel-geoporphyrins moieties in low-grade metamorphic rocks, preserved in situ within cells of a ~1 Gyr-old multicellular eukaryote, *Arctacellularia tetragonala*. We identify these moieties as chlorophyll derivatives, indicating that *A. tetragonala* was a phototrophic eukaryote, one of the first unambiguous algae. This new approach, applicable to overmature rocks, creates a strong new proxy to understand the evolution of phototrophy and diversification of early ecosystems.

[1] Early Life Traces & Evolution-Astrobiology, UR Astrobiology, University of Liège, Liège, Belgium. [2] Commission for the Geological Map of the World, Paris, France. [3] Paul Scherrer Institut, Swiss Light Source, CH-5232 Villigen PSI, Switzerland. [4] Synchrotron Soleil, Saint-Aubin BP 48, France. [5] Unité Matériaux et Transformations (UMR CNRS 8207), Université Lille 1 - Sciences et Technologies, Villeneuve d'Ascq, France. [6] Functional and Evolutive Morphology, Department of Biology, Ecology and Evolution, UR FOCUS, and Center for Applied Research and Education in Microscopy (CAREM-ULiege), University of Liège, Liège, Belgium. [7] Geodynamics & Mineral Resources Service, Royal Museum for Central Africa, Tervuren, Belgium. [8] Research School of Earth Sciences, The Australian National University, Canberra ACT, Australia. ✉email: mcsforna@uliege.be; ej.javaux@uliege.be

Detecting metabolic byproducts in situ in individual microfossils is the key for direct identification of their metabolisms, but up to now, has remained elusive. Among these molecules, tetrapyrroles, such as chlorophylls and hemes, are essential constituents of cellular metabolisms. Whereas these molecules commonly degrade in the early phases of burial and diagenesis[1], they may transform into geoporphyrins under favorable conditions. Geoporphyrins are common in bulk solvent extracts of Phanerozoic sedimentary rocks but exceedingly rare in the Precambrian. The oldest unambiguous occurrence dates back to ~1 billion years (Gyrs)[2]. However, current approaches do not allow the association of the detected porphyrins to individual fossils. In addition, such analyses cannot be performed on overmature rocks because even relatively mild thermal alteration around 200 °C is incompatible with the preservation of free biomarkers[3,4]. Here, we report bound nickel-tetrapyrrole moieties in low-grade metamorphic rocks, preserved in situ within cells of *Arctacellularia tetragonala*, a ~1 Gyr-old multicellular eukaryote from the Congo basin. Combining morphological, chemical, and ultrastructural analyses with synchrotron-based X-ray Fluorescence (SR-XRF) and X-ray Absorption Spectroscopy (SR-XAS), we identify the tetrapyrrole moieties as chlorophyll derivatives, and *A. tetragonala* as one of the earliest multicellular algae. This new methodology, applicable to billion-of-year old, overmature rocks, provides new constraints on the evolution of eukaryotic phototrophy during the Precambrian and the diversification of primary producers in early ecosystems.

We report specific Ni-enrichment in the condensed cytoplasm of cells within several specimens of the organic-walled microfossil *A. tetragonala*. The fossils are preserved as carbonaceous compressions in shales from the shallow marine BII Group in the Mbuji-Mayi Supergroup (Congo Basin, the Democratic Republic of the Congo, Supplementary Information, Supplementary Figs. 1, 2). This sedimentary succession is constrained between 950 and 1030 Ma[5]. Recent Raman geothermometry analyses of the Mbuji-Mayi Supergroup showed that it underwent low-grade metamorphism with a maximal burial temperature of ~200 °C[6]. While far above the oil window and thus possible preservation of biomarkers, previous paleontological investigation of the BII Group reported a large diversity of exquisitely preserved organic-walled microfossils including *A. tetragonala*, comprising 11 unambiguous eukaryotes, 10 possible eukaryotes, and 28 probable prokaryotes[7].

## Results

### Morphology and ultrastructure of *A. tetragonala*.

*A. tetragonala* is ubiquitously found in coeval sedimentary deposits from the Late Mesoproterozoic to early Neoproterozoic from Siberia, Canada, West and Central Africa, China, and India e.g., ref. [7] (for review see Supplementary Table 1). *A. tetragonala* is a readily identifiable microfossil, consisting of unsheathed chains (filaments) of barrel-shaped cells, with deep constrictions between the cells and terminal lanceolate folds at cell ends (Fig. 1 and Supplementary Fig. 2). In the Mbuji-Mayi population we studied (n = 56, Supplementary Fig. 2), fragments of uniseriate filaments (1 to 25 cells) may reach up to 550 μm in length. Cells within these filaments have a similar width (25–45 μm) but variable length (15–100 μm). *A. tetragonala* specimens from the Mbuji-Mayi Supergroup also display a high number of intracellular inclusions (ICI) (Fig. 1b, d, h, i). These structures are organic and show the same thermal maturity as the surrounding walls (Supplementary Fig. 3 and Supplementary Table 2). In addition, these intracellular structures have a greater thickness than the surrounding material, and their shapes, size, and flattening plan reflect the ones of the surrounding cells. These characteristics

indicate that these ICIs, although taphonomically modified, are syngenetic with the fossils and existed before the diagenetic compaction of the fossils[8]. ICIs are common features in shale-hosted assemblages throughout the Proterozoic and the Phanerozoic[7,9–19]. Despite an intense debate that occurred since their first description in Proterozoic microfossils[20], a consensus on their biologically derived nature has been reached for organic-walled microfossils in shales[8,21]. These ICIs are interpreted as condensed cellular material remains (cytoplasm, organelles, and nuclei), because, unlike fossil nuclei, they display an elongated shape following the cells' long axis[8,22]. They thus represent key constituents to investigate possible metabolic traces of these fossils. In the population we studied, the presence of long cells (up to 100 μm, Supplementary Fig. 2) with long ICI within multi-cellular filaments (Figs. 2, 3 and Supplementary Fig. 2) suggests that *A. tetragonala* had a siphonocladous (multicellular multi-nucleate) organization[23]. Some cells may show cross walls evidencing an intercalary division pattern (Figs. 1i, 2a). We also report newly discovered branching patterns in specimens of the Mbuji-Mayi population that were previously unknown for this taxon (Fig.1 and Supplementary Fig. 2). The branching is dichotomous, with equal-diameter branches originating from a nodal cell that is generally larger than the cells within branches (sometimes much larger, Fig. 1c). The nodal cells are rarely preserved (1 to 2 for every 20 specimens), whereas single branches are more common. In some specimens, the three branches are still attached in a Y shape (Fig. 1c, f), suggesting the central nodal cell is not a holdfast structure. We interpret this branching as a true branching system, as opposed to false branching occurring in some sheathed cyanobacteria or algae where cell rows are not connected. These new observations do not justify at this stage the description of a new species since our specimens might represent more complete fragments of the same taxon reported in other successions.

The presence of inward and outward thick lanceolate folds due to compression and the absence of sinuous folds on the fossil surface suggests that the cell walls were flexible but semi-rigid (Supplementary Fig. 4). Scanning electron microscopy images in secondary electrons show smooth and unornamented external and internal walls (Supplementary Fig. 4). Ultrastructural analyses by transmission electron microscopy reveal that the cell walls are bilayered, 100-nm-thick, with an electron-dense thin outer layer and a thick electron-lucent inner layer (Supplementary Fig. 5).

Previously, *A. tetragonala* has been tentatively interpreted as heterocyte or akinete of cyanobacterial genus *Gloeotrichia* spp.[24] and as spores of ascomycete *Fractisporonites*[25]. The combination of a complex morphology comprising true dichotomous branching, node cells, multilayered walls, and large coenocytic cells within multicellular filaments indicates that *A. tetragonala* was eukaryotic, as such combination of characters is unknown in prokaryotes[26–28]. Based on these observations, the hypothesis of a *Gloetrichia*–like cyanobacteria[29] is precluded. Conversely, a branching multicellular filamentous morphology is known in various clades of planktonic and benthic algae within the Stramenopiles and the Archaeplastida[30] (Rhodophyta and Chloroplastida). An ascomycetan affinity[25] cannot be excluded based on morphological features alone. However, no branching has been reported for *Fractisporonites*[31]. Belonging to other ascomycetes is also dubious because spores are usually comprised within an ascus, an outer sheath, which is not observed in our specimens[32].

### Evidence of preservation of Ni-tetrapyrrole moieties in *A. tetragonala*.

To test the biological affinity of *A. tetragonala*, we performed synchrotron radiation-based X-ray fluorescence (SR-μXRF) on 15 extracted specimens (163 cells in total). We

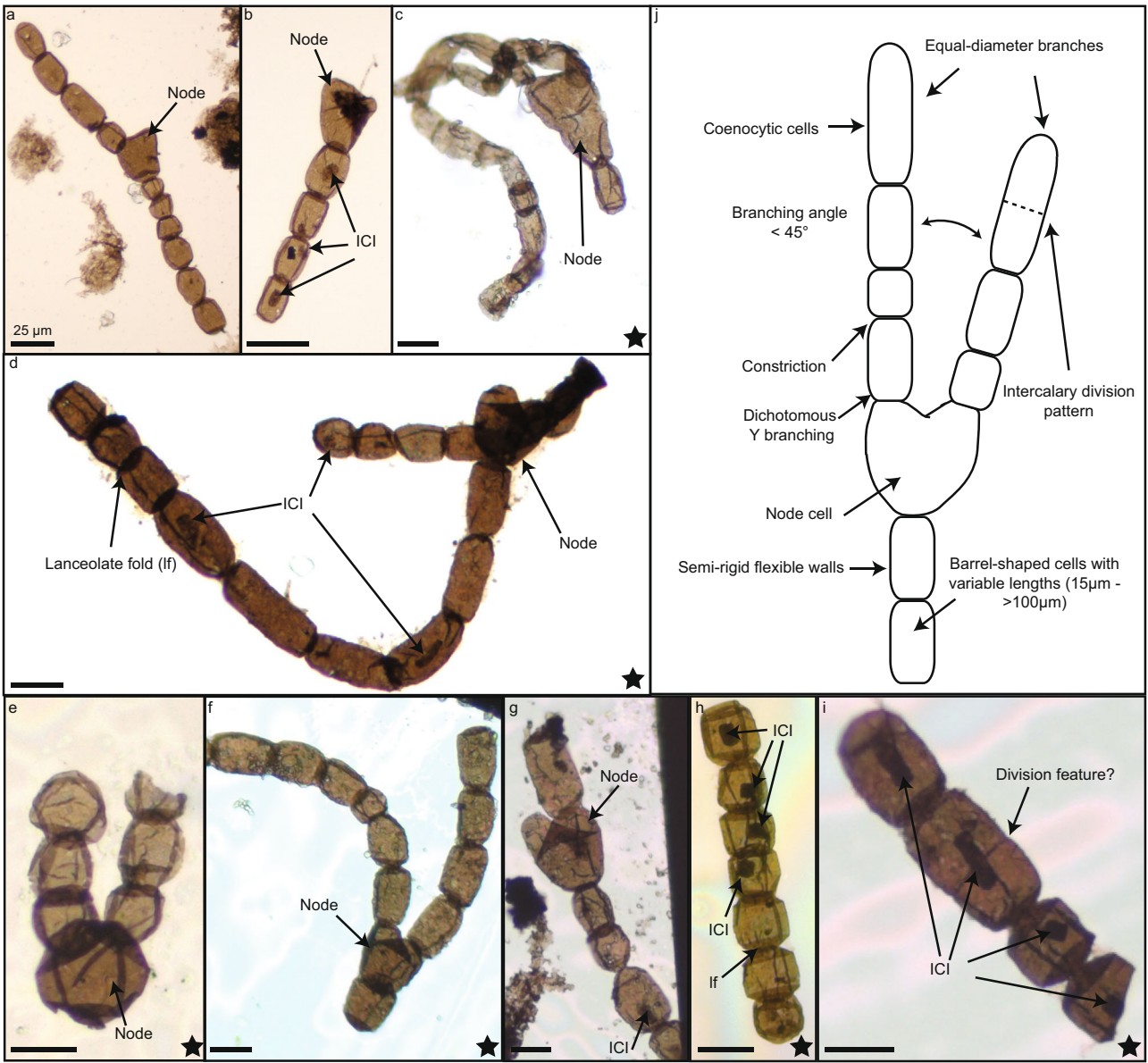

**Fig. 1 Microphotographs of *A. tetragonala* specimens. a–g** Branched specimens with node cell, equal-diameter branches, lanceolate folds (lf), and intracellular inclusions (ICI). **h**, **i** Uniseriate filaments of the microfossils **j**, Sketch of *A. tetragonala*, displaying the main features of the microfossil. Black stars correspond to specimens that have been analysed by SR-μXRF.

show that main metals (Fe, Ni, ±Cu, ±Zn, ±Ca, and ±S) and some minor trace elements (Ti, V, Mn, Cr) are distributed homogeneously in the microfossil walls with a higher concentration in the folds and at the edges of the microfossils due to the substantial thickness of these zones (Fig. 2 and Supplementary Figs. 6–18). Potassium is predominantly associated with small, squared fluorides (Fig. 2, fl.), neo-formed during drying of the storing solution after demineralization. Calcium is always found in these fluorides, and Fe, Ni, Cu, and Zn are occasionally present. Metals can also form small hotspots attached to the microfossil surfaces, possibly in oxides or sulfides (Fig. 2 and Supplementary Figs. 6–18, s.).

The ICIs are specifically enriched in Ni compared to the walls (1.5 to 17 times more Ni in the ICIs than in the walls, Supplementary Table 2) whereas Ni in the walls shows the same distribution patterns than other metals. The Ni-rich ICIs show a limited presence of Fe (Fig. 2). Within different cells of a single fossil fragment, ICIs display different Ni and Fe concentrations.

Some ICIs can be iron-free while other display various Fe enrichment (e.g. Supplementary Fig. 10). High-resolution SR-μXRF shows that when Fe is present, it is not directly associated with Ni but forms small nuggets randomly distributed in the ICI (Fig. 2j). On the contrary, Ni is widely and homogeneously distributed within the ICI.

The origin of Ni in the ICIs could have been endogenic or exogenic. Ni is little used by eukaryotes except by fungi and some plants, which require Ni for their nitrogen cycle[33]. Nickel is transported to sediments by complexation and sedimentation with organic material before being freed in the pore waters during the remineralization of organic material. When in pore waters, Ni becomes available for Ni-sulfide precipitation or for passive diffusion within the organic material[34,35]. The proportion of each phase is controlled by the environmental conditions prevailing, with the formation of Ni-sulfide precipitation enhanced by sulphidic conditions and the incorporation within the organic fraction enhanced in non-sulphidic, anoxic conditions[35]. Raman

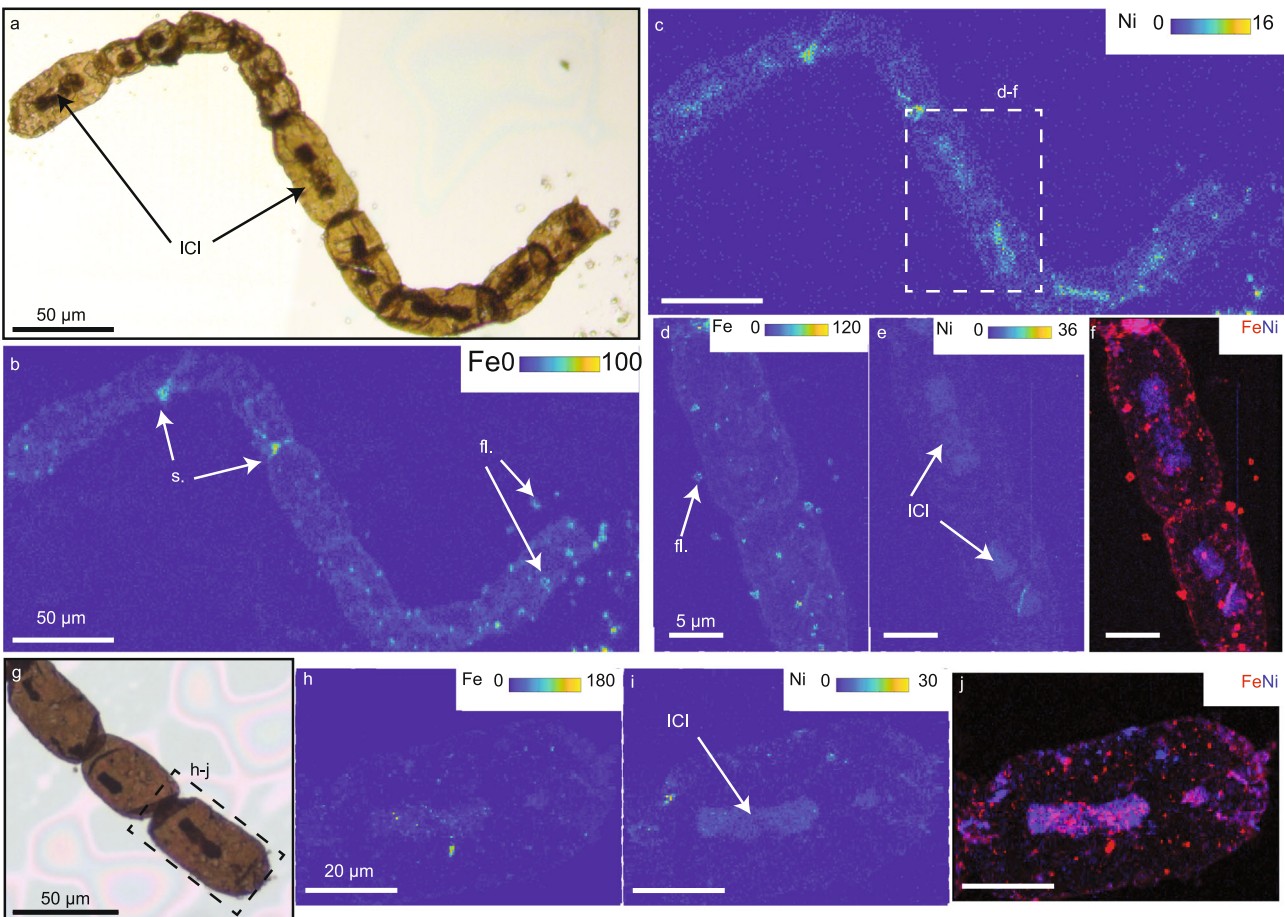

**Fig. 2 Nickel-specific enrichment in intracellular inclusions. a, g** Microphotographs of the studied specimen. **b**–**f** Fe and Ni SR-μXRF maps obtained at SLS (**b**, **c** pixel: 1 μm, 200 ms/px), at SS (**d**, **e** pixel: 400 nm, 100 ms/px) and, the associate composite image (**f**, R: Fe, B: Ni). These maps show a specific enrichment of Ni uncorrelated of Fe in the intracellular inclusions (ICI) while the walls appear homogeneously enriched in Fe and Ni. **h**–**j** Fe and Ni SR-μXRF maps obtained at SLS (**h**, **i** pixel: 1 μm, 1 s/px) and the associated composite image (**j**, R: Fe, B: Ni). These maps show that some ICIs can also be enriched in Fe, but that Fe is present as small hotspots in the ICI while Ni is mainly homogeneously distributed. Color scales correspond to normalized counts. (s.) is for sulfides and (fl.) is for fluorides attached to the surface of the fossils.

spectra do not evidence mineral phases associated with the ICIs (Supplementary Fig. 3) and the Linear Combination Fitting (LCF) of the Ni K-edge XANES spectra exclude a percentage above a few percent of inorganic Ni in the ICIs (Fig. 3, Supplementary Information, Supplementary Fig. 18, and Supplementary Table 3). The Ni K-edge XANES spectra obtained on the ICIs are characterized by a small pre-edge peak at ~8333 eV (A, Fig. 3d), a shouldering at ~8339 eV (B, Fig. 3d), and two peaks at ~8350 and ~8358 eV on the principal edge (C-D, Fig. 3d) while inorganic Ni (NiO, Ni(OH)$_2$, NiCO$_3$, NiSO$_4$(H$_2$O)$_{6-7}$) reference spectra exhibit a strong narrow white band feature centered between 8348 and 8352 eV (Fig. 3 and Supplementary Fig. 18). This suggests that the Ni is associated with organic compounds in *A. tetragonala* ICIs and that diagenesis took place in anoxic, non-sulphidic conditions. Such preservation conditions are also supported by iron speciation data for shales from the Kanshi drillcore[36].

The specific association and enrichment of Ni with the ICIs of *A. tetragonala* suggests that the Ni is bound to ligands that strongly and specifically bind Ni. These ligands largely originated in the cytoplasm (Fig. 2). Naturally occurring, organic Ni-complexes exist as mixed-ligand tetradentates, humate complexes, and tetrapyrrole complexes[34]. Mixed-ligand tetradentate and humate complexes can be excluded as they have relatively low affinities to Ni and are unlikely to persist in sedimentary

environments[34]. By contrast, tetrapyrrole complexes, such as porphyrins, have extremely strong Ni-chelating properties and high stabilities, persisting in geological environments for hundreds of millions of years[2,34]. In fact, the high affinity of tetrapyrroles to bivalent Ni and V appears to be the primary controlling factor for the accumulation of these elements in sedimentary environments and petroleum[34]. While iron, copper, and zinc-porphyrins are also occasionally recovered from petroleum e.g., ref. [37,38], these complexes are significantly less stable[34], potentially explaining the low abundance of these metals in the ICIs.

The accumulation of Ni in *A. tetragonala* within tetrapyrrole complexes is confirmed by XANES data. The overall spectral shape and the positions of the A, B, C, and D bands are congruent with Ni(II) in porphyrinic square-pyramidal N coordination as observed in Ni-porphyrin standards such as Ni-octaethyl porphyrin [NiOEP] and Ni-tetraphenyl porphyrin [NiTPP][39] (Fig. 3d). This is also supported by the LCF, which estimates that more than 90% of the total Ni is in this coordination (Fig. 3d, Supplementary Fig. 18, and Supplementary Table 3). By contrast, the spectra are inconsistent with Ni atoms randomly occupying vacancies in graphitic carbon[40], humates ligands[41], and with nickel coordinated in NiN$_2$O, NiN$_2$S, and NiN$_3$ complexes[42]. When compared to free porphyrins, the XANES spectra of Ni in the ICIs show a

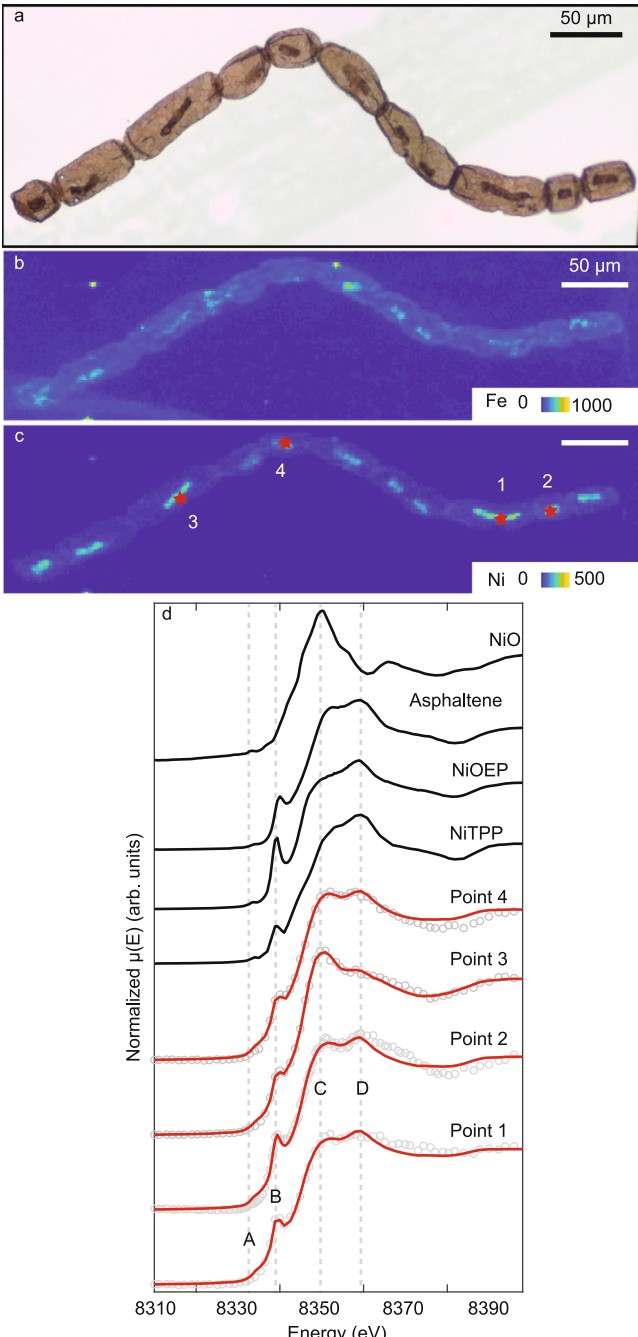

**Fig. 3 Presence of tetrapyrrole moieties highlighted by XANES analyses on intracellular inclusions. a** Microphotograph of the studied specimens. **b**, **c** Fe and Ni SR-μXRF maps obtained at SLS (pixel: 1.5 μm, 200 ms/px) showing the enrichment in Ni and Fe of the intracellular inclusions (ICI). Color scales correspond to normalized counts. **d** XANES spectra at the Ni K-edge of 4 ICIs performed in zones with low Fe content (gray circles) and their linear combination fitting (red lines) and XANES spectra of two Ni-porphyrin (NiTPP: Ni(II)-tetraphenylporphine, NiOEP: Ni(II)-octaethylporphine)[43], asphaltene[44], and NiO standards. The shoulder and the spectral line shape are typical of Ni in coordination (IV) in bound Ni-porphyrinic species. Differences between the fitted spectra and the data come from the molecular heterogeneities between the standards used for the fitting and the incorporated tetrapyrroles moieties in the kerogen.

broadening of some spectral features and minor differences in relative intensities and shouldering positions (Fig. 3d). Such variations have been attributed to the distortion of the coordination of Ni(II) in porphyrins that are bound into a macromolecular network, as observed for example in asphaltenes[39,43,44]. Notably, the typical XANES spectra of covalently bound tetrapyrrole complexes are conserved in coking residues of bitumen that have been heated to temperatures as high as 480 to 565 °C[43], attesting to their extraordinary thermal stability and explaining the preservation of Ni-tetrapyrrole moieties in billion-year-old fossils that experienced prehnite-pumpellyite facies metamorphism.

**Origin of the Ni-tetrapyrrole moieties.** As Ni-tetrapyrrole are common molecules in Phanerozoic petroleum and can be found up to 1.1 Ga[2], it is important to assess the syngenicity of the tetrapyrrole recovered in the microfossils. Oil migration from a late Neoproterozoic/Early Paleozoic active petroleum system has been observed in the Sankuru-Mbuji-Mayi-Lomami-Lovoy[45] and is thought to have occurred during the Cambrian-Ordovician. As a result of this late oil migration, solid asphaltites have been reported in the BIIc dolomites in cracks and as inclusions. However, our Raman spectra indicate that the ICIs experienced the same thermal maturation than the walls (Supplementary Fig. 4 and Supplementary Table 2). In addition, the only reported Raman spectrum by ref. [45] on these asphaltites shows a less mature signature than the kerogen in KN 22 and KN23[45]. This excludes contamination of the ICIs during the oil migration episode. Synsedimentary bitumen were also reported in the Mbuji-Mayi shales but are not associated with the microfossils[6], as observed, for instance, in the Gunflint chert microfossils[46], where bitumen fills the silicified microfossils preserved in 3D. The context of deposition and preservation in shales (carbonaceous fossils compressed in 2D parallel to the bedding with continuous walls) is indeed less favorable for bitumen migration than in chert where the bitumen migrates through the silicifying microfossils and fills semi-hollow molds[46]. Moreover, the distribution of ICI relative to the fossils is non-random, on the contrary to migrating oil. The presence of a single ICI inside cells (only 1 per cell), their absence on the outer fossil walls, their nearly constant shape, and their occurrence in only a few species in the assemblage (including *A. tetragonala*[7]), and not all specimens of these species, exclude a bitumen origin. In *A. tetragonala*, the tetrapyrrole moieties are specifically associated with the ICIs and not with the walls (either from ICI-containing or empty cell walls) as showed by the Ni distribution in the cells. As these ICIs are considered structures forming before or during burial[8], the Ni-tetrapyrrole moieties are syngenetic with these ICIs, originating from the living organisms and not from late contamination during oil migration.

Ni-porphyrins are the diagenetic products of biological tetrapyrroles, predominantly chlorophylls and hemes[34,47]. Tetrapyrroles are pivotal constituents of the cell metabolism in all three domains of life[48]. Hemes are involved as redox center in electron transport chains, as a cofactor of numerous proteins but also in regulatory processes as sensors[48]. In phototrophs, chlorophylls are the main photosynthetic pigments and represent most of the tetrapyrroles present in the cells[48]. In algae, the concentration of chlorophyll[49–51] can be extremely high, reaching between 1.6 and 17 mM (Supplementary Table 5). By contrast, cellular heme concentrations in algae are commonly 200 to 400 times lower than chlorophylls[49], ranging between 4 and 72 μM. Cellular heme

concentrations in yeast are lower yet[52–54], ranging between 0.5 and 5 μM (Supplementary Table 5). Based on SR-XRF mapping, we estimated Ni concentrations in the ICIs between 13 and 23 ng cm$^{-2}$, corresponding to minimal cellular tetrapyrrole concentrations between 25 and 43 μM (Supplementary Table 6). The tetrapyrrole abundances in fossil *A. tetragonala* cells thus exceed the concentration of heme in living yeast[52–54] (Supplementary Tables 5, 6), excluding a fungal affinity. By contrast, considering a likely order of magnitude loss of pigments during diagenesis and metamorphism[1], the computed tetrapyrrole concentrations compare favorably with the tetrapyrrole heme +chlorophyll content of algae[49–51] (Supplementary Tables 5, 6). The combined fossil and tetrapyrrole evidence thus suggests that *A. tetragonala* was a photosynthetic eukaryote, a multicellular branching alga member of the Archaeplastida (Supplementary Information), an affiliation consistent with most molecular clock estimates and the fossil record e.g., refs. [55–60].

While free porphyrins are only preserved in thermally exceptionally well-preserved bitumens and are extremely rare in the Precambrian, XANES of metal-bound molecules within fossil cells offers a new methodology that can reveal degraded porphyrins in organic matter metamorphosed to prehnite-pumpellyite facies. This approach offers the possibility to track porphyrins, and thus phototrophy, much further back in time, perhaps even into the Archean. Indeed, the ability of XANES to detect tetrapyrrole structures bound to insoluble organic matter reduces the risk of contamination and allows the assignment of bound porphyrin derivatives to individual microfossils. ICIs are common in a large variety of prokaryotic and eukaryotic microfossils, and our results demonstrate that in situ detection of key metabolism byproducts should be possible in ICIs within low-grade metamorphic rocks, providing a new complementary approach to decipher the biological identity of individual, enigmatic microfossils, and opening a new window into early Earth ecosystems.

## Methods

**Sample preparation—Extraction of microfossils**. The *Arctacellularia* microfossils were extracted from shales following a modified preparation procedure described by ([61]), avoiding centrifugation and mechanical shocks that could damage the fossilized forms and oxidation that could alter the kerogenous wall chemistry and color. After cleaning, the samples were crushed in a mortar. Carbonates were removed by hydrochloric acid (HCl, 35%) and silicates by hydrofluoric acid (HF, 60%). Neo-formed fluorides were removed by hot HCl. The organic fraction was filtered by hand. Extracted kerogen was stored in millipore water. Fifty-six specimens of *A. tetragonala* were studied here, 43 with optical microscopy, 15 with SR-μXRF, 1 with Raman spectroscopy, 7 with SEM, and 1 with TEM.

**Scanning electron microscopy (SEM)**. Before the experiment, microfossils were pipetted under an inverted microscope and deposited on a silicon wafer, and air-dried. SEM images were collected at the Centre commun de microscopie de Lille (Unité Matériaux et Transformations, University of Lille, France) on carbon coated microfossils in second electron mode using a JEOL JSM-7800F LV (JEOL). Analytical conditions were 5 kV-accelerating voltage, low current, and a 10.1 mm working distance.

**Transmission electron microscopy (TEM)**. To characterize their ultrastructure, extracted microfossils were embedded in agarose (1%) before being progressively dehydrated in gradual series of ethanol baths (30, 50, 70, 90, and 100%) and gradually impregnated with hard grade LR white resin. Ultrathin sections were made at the CAREM-University of Liege (Center for Applied for Research and Education in Microscopy) with a Reichert Ultracut E ultramicrotome using a diamond knife and mounted on Formvar-coated copper grids (400 mesh). Observations were performed on a TEM FEI TITAN THEMIS 300 (Centre commun de microscopie de Lille (Unité Matériaux et Transformations, University of Lille, France) working at 300 kV-accelerating voltage and on a TEM/STEM Tecnai G2 Twin (CAREM-Ulière) working at 200 kV-accelerating voltage.

**Synchrotron radiation X-ray microfluorescence (SR-μXRF)**. Before the experiment, microfossils were pipetted under an inverted microscope and dropped on

silicon nitride (Si$_3$N$_4$) windows. The windows were air-dried. The distribution of metals was characterized by SR-μXRF during two beamtimes at the Swiss Light Source (SLS; microXAS-X05LA beamline, Villigen, Switzerland) and during two beamtimes at the Synchrotron Soleil (SS; Nanoscopium beamline, Gif-sur-Yvette, France). During the first experiment at SLS (May 2018), maps were acquired with a 1 μm² spatial resolution. Two acquisition modes were used, the Stop-and-Go mode where a full XRF spectrum is acquired for each pixel, and the On-the-Fly mode were maps for each element correspond to the integration of the counts in the pre-defined Region of Interest (ROI). The 16.2 keV excitation energy allowed the detection of elements between Si and Sr (K lines) and Ba and Bi (L lines). During the second experiment at SLS (May 2019), the maps were acquired with the same experimental conditions but with an 8.4 keV excitation energy. This allowed the detection of elements between Si and Ni (K lines). For both experiments at SS (July 2018, May 2019), the maps were acquired with pixels between 300 and 500 nm² and with a 16.1 keV excitation energy, allowing the detection of elements between Si and Rb (K lines) and between Ba and Pb (L lines).

Spectral intensities were normalized by argon counts recorded for each pixel to prevent the variation of concentration to be linked with a variation of the incoming photon flux. We were thus able to compare spectra and distribution maps acquired on both synchrotrons (i.e., using different detectors and experimental setups). XRF spectra were treated with PyMca and Matlab software. Ni/Fe ratios in walls and in ICIs were obtained by calculating the ratio of the peak areas of each considered element, after normalizing the sumspectra of subzones in the map by the total integration time of each map (time of analysis per pixel*number of pixel per map)[62].

**X-ray absorption spectroscopy**. X-ray Absorption Near-Edge Structure (XANES) Spectroscopy was performed at the Swiss Light Source on the microXAS-X05LA beamline (May 2019). The incident X-ray energy was selected using a Si (1 1 1) double crystal monochromator. The beamspot size was focused to ~1 μm. Analyses were performed punctually on the Ni-rich spot identified with SR-XRF mapping. Replicate spectra ($n = 4$ to 6) for each analysis point were collected in fluorescence mode by recording the Ni Kα (8.33 keV) and tuning the monochromator energy from 8.31 to 8.42 keV. Energy calibration was performed with respect to the K-edge of a Nickel (0) foil (8333 keV). Replicate transmission XANES spectra for reference compounds (Ni(OH)$_2$, NiO, NiCO$_3$, NiSO$_4$(H$_2$O)$_{6-7}$) were also recorded. Ni-porphyrin standards for Ni-octaethyl porphyrin (NiOEP), Ni-tetraphenyl porphyrin (NiTPP), and asphaltene standard were taken from (43, 44). Energy calibration, pre-edge background subtraction, and post-edge normalization were performed prior to linear combination fitting (LCF) with the Athena® software[63]. The fitting range ranged from −20 to +20 eV relative to the theoretical Ni K-edge energy. LCF was performed with no constrained individual component weights nor their weight sum following (43). The assessment of the quality of the fitting was done by looking at the smallest R-factor and the combination of components. Results of the LCF can be found in Fig. 3, Supplementary Fig. 18, and Supplementary Table 3.

**Estimation of minimum tetrapyrrole concentrations in living *A. tetragonala* cells**. To constrain the origin of the tetrapyrrole moieties preserved in the microfossils, we computed the concentration of tetrapyrrole from the Ni area densities measured in the SR-XRF distribution maps in the *A. tetragonala* from Fig. 3. We considered that each Ni atom in the collapsed ICI of the fossil cells corresponds to one tetrapyrrole moiety. Parameters and results of the calculation can be found in Supplementary Table 3. The computed cellular tetrapyrrole concentrations in the fossils are minima because the original pigments were presumably attenuated by one or two orders of magnitude during diagenesis and metamorphism. Taking such loss into account, we compare these tetrapyrrole estimates to heme and chlorophyll concentrations in algae and fungi from the literature (Supplementary Tables 4 and 5).

To compute tetrapyrrole concentration from Ni abundance, we considered the four ICIs where the XANES analysis was performed in Fig. 3. The first step was to estimate the Ni area density for each considered ICIs. For that, we calculated the mean Ni count rates in the ICIs from the Ni SR-XRF distribution map (Supplementary Table 3). Applying identical experimental conditions when acquiring the SR-XRF maps in May 2019 at SLS, we performed measurements on a nickel calibration foil with a known area density of 2.32 μg cm$^{-2}$ and obtained a signal intensity of 28,000 counts per second. By comparing this standard to the mean count rates obtained for the considered ICIs, ranging from 156 to 278 counts per second, we obtained Ni area densities ranging from 13 to 23 ng cm$^{-2}$ (Supplementary Table 3).

From the area density of nickel in the ICI, it is then possible to compute the mass of nickel and then its number of mole in a single cell, considering that the projected area of the ICI corresponds to 10% of the projected area of the cell. Following Eq. (1), we obtain a mass of nickel per cell between 0.97 $10^{-14}$ and 4.6 $10^{-14}$ g (Supplementary Table 3), corresponding to a number of mole between 1.7 $10^{-16}$ and 7.9 $10^{-16}$ mol (Eq. (2), Supplementary Table 3).

$$m_{Ni} = (AD_{Ni})_{ICI}.0.1A_p \qquad (1)$$

With $m_{Ni}$ the calculated mass of nickel, $(AD_{Ni})_{ICI}$ the area density of nickel in the

ICI, and $A_P$ the projected area of the cell

$$n_{Ni} = \frac{m_{Ni}}{M_{Ni}} \quad (2)$$

With $n_{Ni}$ the number of mole of Ni, $m_{Ni}$ the calculated mass of nickel, and $M_{Ni}$ the molar mass of Ni ($58.69\,g\,mol^{-1}$).

We then estimated the volume of *A. tetragonala* cells, considering that *A. tetragonala* cells were cylindrical. As cells undergo a flattening during fossilization, their apparent radii increase while their lengths stay the same[64]. Therefore, to obtain the volume of the living cell, we need to apply a morphometric correction factor to the measured radii ($R_m$) as for cylindrical-septate filaments[64] following the Eq. (3). The volume of the living cell can then be calculated with the Eq. (4):

$$R_c = \frac{2}{\pi} \cdot R_m \approx 0.64 \cdot R_m \quad (3)$$

with $R_c$ the corrected radius, $R_m$ the measured radius

$$V_{cell} = L.\pi.R_c^2 \quad (4)$$

with $V_{cell}$ the calculated volume of the living cell, $L$ the measured length of the cell, $R_c$ the corrected radius.

From the estimated volumes (Supplementary Table 4), we infer a minimum cellular tetrapyrrole concentration of 25 to 43 µM (Supplementary Table 4) (5).

$$C_{Ni} = C_{tetrapyrrole} = \frac{n_{Ni}}{V_{cell}} \quad (5)$$

With $C_{Ni}$ the molar concentration of Ni equivalent to $C_{tetrapyrrole}$, $n_{Ni}$ the molar concentration of tetrapyrrole and $V_{cell}$ the volume of the considered cell.

The minimum concentration of tetrapyrroles of 25 to 43 µM preserved in the fossil cells are higher by a factor of 10 to 100 to those detected in living yeast[52–54] (0.5–5 µM; Supplementary Table 5), making fungal heme an unlikely source for the Ni tetrapyrroles moieties in the fossils. However, considering that the bulk of the tetrapyrroles must have decayed after cell death and during burial and metamorphism[34], the values are in agreement with chlorophyll concentration found in modern green algae[49–51] (1.6 to 17 mM, Supplementary Table 5). The differences between our fossils and fresh algal cells might result from the loss of biomolecules during diagenesis.

**Raman spectroscopy**. The mineralogy of the sample and the kerogen thermal maturity were investigated by Raman spectroscopy at the University of Liège (Early life Traces & Evolution—Astrobiology Laboratory, Belgium). Raman spectra were collected on polished freestanding thin section and on freestanding extracted microfossils deposited on ZnSe plates using a Renishaw Invia Raman microspectrometer with an Ar-ion-40 mW monochromatic 514 nm laser source. The laser excitation was focused through a 100 x objective to obtain a 1–2 µm spot size. To limit the damages on the microfossils, spectra were acquired with a laser power lower than or equal to 10% and an integration time of 10x1s. Acquisitions were obtained with a 1800 l/mm grating illuminating 1040 × 256 pixel CCD array detector. Spectra were acquired in static mode fixed at 1150 cm$^{-1}$, allowing a 2000 cm$^{-1}$ detection range and a 1 cm$^{-1}$ spectral resolution. Beam centering and Raman spectra calibration were performed daily on a Si-glass with a characteristic Si-band at 520.4 cm$^{-1}$. Spectra were processed with Wire 4.2® software (Renishaw). Details on the acquisition parameters, data treatment, and geothermometry can be found in ref. [6]. To assess correctly the nature and the syngenicity of ICIs, spectra were acquired on the fossil walls and the ICIs.

## Data availability

The authors declare that data supporting the findings of this study are available within the paper and its supplementary information files. The fossil material is stored in the collections of the Early Life Traces & Evolution—Astrobiology Laboratory, University of Liège, Belgium. Correspondence and requests for materials should be addressed to M.C.S. (mcsforna@uliege.be) or E.J.J. (ej.javaux@uliege.be).

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

## Acknowledgements

We thank the Royal Museum for Central Africa (Tervuren, Belgium) for access to the Kanshi SB13 drillcore. We thank M. Giraldo, A. Lambion at the Early Life Traces & Evolution—Astrobiology lab, and S. Smeets at the CAREM platform (University of Liège, Belgium) for technical support. The authors acknowledge the Paul Scherrer Institute, Villigen, Switzerland, and the Synchrotron Soleil, Gif-sur-Yvette for provision of synchrotron radiation beamtimes at microXAS-X05LA and the Nanoscopium beamlines, respectively. We thank M.B.J. Lindsay (University of Saskatchewan, Canada) for providing XANES porphyrins spectra and P. Cardol (University of Liege, Belgium) for information on chlorophyll concentration in modern algae. The Marie-Curie Cofund program at the University of Liège and the FNRS CR PROMESS project (M.C.S.), the FRS-FNRS-FWO EOS ET-Home grant 30442502 (E.J.J.), the ERC Stg ELiTE FP7/308074 (E.J.J.), the BELSPO BRAIN project B2/212/PI/PORTAL (E.J.J., Y.J.L.), and the program CALYPSO PLUS (Synchrotron Soleil) (M.C.S., C.F.D., C.C.L.) supported this project.

## Author contributions

Conceptualization: M.C.S, E.J.J. Investigation: D.B. provided access to samples and geological information. M.C.S., C.C.L., C.F, C.F.D., Y.C., E.J.J., D.G., D.F.S., K.M., A.S. collected the SR-µXRF and XANES data. M.C.S., C.C.L., C.F.D., C.F. with inputs of Y.C. and Y.J.L performed the FT-IR and Raman analyses. M.C.S., C.F.D., C.C.L., A.A., A.F., P.C. made SEM and TEM observations. M.C.S., J.J.B, D.G., C.F.D. performed tetrapyrrole estimation calculations. Funding acquisition: E.J.J., M.C.S. Writing – original draft: M.C.S., C.C.L., E.J.J., J.J.B. Writing – review & editing: C.F.D., C.F., Y.C., Y.J.L., D.G., D.F.S., K.M., A.S., A.A., A.F., P.C., D.B.

## Competing interests

The authors declare no competing interests.
