## [Peer review file · Nature Communications]

Intracellular bound chlorophyll residues identify 1 Gyr-old fossils as eukaryotic algaeReviewers' comments:

Reviewer #1 (Remarks to the Author):

This manuscript describes the application of microscopy, SR-uXRF, and Raman spectroscopy to a total of 56 microfossils. These techniques reveal the presence of inclusions inside microfossils that appear to be composed of Ni-porphyrins. The manuscript describes the morphology of the microfossils as suggestive of algae, the XANES data as indicating the presence of Ni-tetrapyrroles, indicating chlorophyll, and thus argues that the inclusions indicate that these are microfossils of algae that contained chlorophyll. This method is also proposed as being useful for identifying metabolic products in other fossils.

I have several concerns about the manuscript in its present form:

First, the paper never demonstrates that the ICI are syngenetic with the microfossils, rather than things introduced later by diagenetic processes. These inclusions are described as "condensed cytoplasm" but it is never explained how that is a known fact. However, abelsonite, mineral deposits of Ni-porphyrin, are known to form in vugs and fractures as a secondary mineral, as a result of migration (e.g., Mason et al., 1989), which would seem to provide a secondary explanation for these inclusions. Additionally, the analyses of the ICI are complicated by the fact that there are not analyses of the matrix. The manuscript notes that Ni-porphyrin complexes are accumulated in sedimentary environments (131) so it is especially important to establish that these ICI features originate from the fossilized organism, and not from diagenetic process. Being able to compare an inside-the-fossil signal with an outside-the-fossil signal would help, but the paper only describes isolated microfossils.

The identification of Arctacellularia as algae is also problematic, as it seems to hinge on the morphology of the samples and calculations of estimated Chl content from SR-XRF mapping. The manuscript indicates that these microfossils were previously identified as cyanobacterial or fungal (line 79-80), then describes the morphology of these microfossils as having branching patterns are "previously unknown" for this taxon (line 63-64). However, it also concludes that these fossils cannot be fungal as branching has not been reported for that particular microfossil (line 87-88). This seems contradictory, especially as later the manuscript notes that fungus sequester Ni as part of their metabolic process (106-107). Later in the manuscript, the concentration of Ni is used to estimate how much heme and chlorophyll these organisms would have contained, indicating that this concentration suggests that this microfossil is algal. However, this calculation relies on several assumptions, many unstated, including that each Ni molecule can be directly correlated with a tetrapyrrolic molecule.

There are other examples of unclear or incomplete data analysis. For instance, when the XANES data are described, the relevant figures show iron sulfide complexes on and in the microfossils (i.e., Fig 2b has sulfide labeled, and the supplementary figures show the co-occurrence of iron and sulfur) but the manuscript describes diagenesis occurring in a "non-sulphidic" condition (121). Similarly, although the Raman spectrum is described as showing no evidence for inorganic Ni compounds inside an ICI (Fig S18, line 113-114), it also shows no indication of any Ni-pyrrole compounds. The spectrum shown is labeled as containing D and G carbon bands, there are no bands indicative of a Ni-porphyrin, complexes that have been studied thoroughly by Raman spectroscopy (i.e., Schindler et al., 2018, Ruff data base of Raman spectra of minerals). If these ICI are composed of Ni-porphyrin the fact that Raman spectroscopy does not contain evidence of this composition should be discussed. Finally, with regards to data analysis and interpretation, the XANES data are described as being "typical of Ni in coordination (IV) in bound N-porphyrinic species" (Line 325) but no line fitting is ever done, instead there is just a stack of spectra to eyeball.

Although this combination of methodologies is intriguing, it is not yet convincing that this is a robust new path forward. In addition to the concerns listed above, the manuscript never describes what other metabolic compounds could be identified with this approach.

Reviewer #2 (Remarks to the Author):

Sforna et al. demonstrate in this short communication that intracellular inclusions in the fossil *A. tetragonala* contain Ni-porphyrins likely originating from chlorophyll, thus providing compelling and direct evidence that this organism was a photosynthetic eukaryote.

I consider this an elegant and quite interesting piece of work that, among other things, adds substantial weight to the notion that crown group eukaryotes were already quite diverse by 1.0 Ga... in particular, given that recently there has been somewhat of a case made for a very late radiation of the group. The authors tease the possibility of this approach being applied to much older rocks of Archean age, which I do find quite exciting.

I find the data SR- μ XRF and XANES data particularly convincing, although I am not myself a practitioner of these methods. Materials and methods are well described in the supplements. I have no major issues with the work. Just some minor comments:

Line 39, 164. Wouldn't it be better to exchange here "phototrophy" for "photosynthetic eukaryotes"? It is just that phototrophy is quite a broad term and in this particular case you can be much more specific.

In your first and second paragraph, it is sometimes not clear what data is coming from the current work and what data you are citing from previous work as part of introduction and providing context. Could you please just make sure that you delineate and separate better the background work leading to this paper from the new data being reported?

Lines 60 to 61. It is unclear whether this interpretation (e.g. multinucleate) is based on refs. 8 and 9 cited in the sentence before, or something that you deduce from the current data. Please reference this better or elaborate on the evidence. In any case, are you implying that the ICI is therefore residue from the nucleus? I bring this up because you then go to demonstrate that ICI contains a rather uniform distribution of Ni-tetrapyrroles. In addition, you make a case for the concentrations of these chlorophyll by-products to have decreased by orders of magnitude during diagenesis. Does not this indicate that ICI is more likely to be originating from chloroplast material? Or at least, in this particular case, to also contain substantial chloroplast remnants? Perhaps, you should state this explicitly in the text and what this would imply for other eukaryotic fossils that contain ICI. I am thinking, in particular, of those 1.6 Ga fossils by Bengtson et al. (Plos Biology 2017), who I believe make a case for pyrenoid remnants, which are structures associated with chloroplasts...

Finally, I think the more broader audience interested in the subject would benefit from having access to a summary of the evidence placing *A. tetragonala* within Archeoplastida, mentioning whether this is considered to be from a lineage with no easily identifiable living representatives or if it has been placed within a known group. It would also be useful if you can highlight whether there is any conflicting interpretations of fossils identified as *A. tetragonala*. Just adding a paragraph into the Supplementary Information on this, perhaps associated with your Extended Data Table 1, would be nice.

Answer to reviewers

Reviewer #1 (Remarks to the Author):

This manuscript describes the application of microscopy, SR-uXRF, and Raman spectroscopy to a total of 56 microfossils. These techniques reveal the presence of inclusions inside microfossils that appear to be composed of Ni-porphyrins. The manuscript describes the morphology of the microfossils as suggestive of algae, the XANES data as indicating the presence of Ni-tetrapyrroles, indicating chlorophyll, and thus argues that the inclusions indicate that these are microfossils of algae that contained chlorophyll. This method is also proposed as being useful for identifying metabolic products in other fossils.

I have several concerns about the manuscript in its present form:

First, the paper never demonstrates that the ICI are syngenetic with the microfossils, rather than things introduced later by diagenetic processes. These inclusions are described as condensed cytoplasm never explained how that is a known fact.

The ICIs from the 65 fossils we studied for this article all answer to the criteria given by Pang et al (2013) to insure that the ICIs are syngenetic and existed before the post-mortem collapse or diagenetic compaction of the vesicle walls. Pang et al. (2013) showed that although ICIs are taphonomically modified (fusion with the cell walls), ICIs existed before the post-mortem collapse or diagenetic compaction of the vesicle walls if the ICIs meet following conditions:

- 1) the size and the shapes of the ICIs reflect the appearance of the surrounding cell
- 2) The ICIs and the enclosing cells show the same answer to any stress field during postmortem of diagenetic deformation
- 3) The ICIs have a greater thickness than the surrounding material and marked by a considerable, granular texture, whereas the rest of the cell appears smoother and flatter.

A large bibliography on Intracellular inclusions (ICIs) does exist and similar inclusions have been documented for many Proterozoic and younger organic-walled microfossils over more than 50 years of palynological studies. Despite an intense debate occurred since their first description in Proterozoic microfossils during the sixties, a consensus on their biologically-derived nature has been reached for organic-walled microfossils in shales (see Pang et al., 2013). ICIs are well-known features occurring in Proterozoic and Phanerozoic organic-walled microfossils both in cherts and shales (e.g. Barghoorn and Schopf, 1965; Pang et al., 2013; Loron and Moczydlowska, 2017; Adam et al., 2017; Baludikay et al., 2016; Beghin et al., 2017; Leiming et al., 2005; Tang et al., 2013; Agic et al., 2017; Grey, 2005; Grey et al., 2003; Prasad et al., 2005; Bengston et al., 2017; Marshall et al., 2017; Wellman et al., 2019; Nowak et al., 2017; etc.). Different origins for these intracellular inclusions have been proposed including diagenetic

carbonaceous concretions, taphonomically or biologically condensed cytoplasm, nuclei, mitochondria, chloroplast, pyrenoids...(e.g. Oehler, 1977, Niklas and Brown, 1981, Dejax et al., 2001, Pang et al., 2013; Knoll & Barghoorn, 1975; Li et al., 2012; Tang et al., 2013; Bengston et al., 2017; Nowak et al., 2017; Moczydlowska, 2016; Carlisle et al., 2021).

To clarify the affinity and syngenicity of these ICIs we accordingly modified the manuscript.

“A. tetragonala specimens from the Mbuji-Mayi Supergroup also display a high number of intracellular inclusions (ICI) (Fig.1b, d, h, i). These structures are organic and show the same thermal maturity than the surrounding walls (Supplementary Fig. 3), with greater thickness than the surrounding material. Their shapes and size reflect the ones of the surrounding cells. These characteristics indicate that these ICIs, although taphonomically modified, existed before the diagenetic compaction of the fossils⁸. These ICIs are interpreted as condensed cellular content remains (cytoplasm, organelles and nuclei), because, unlike fossil nuclei, they display an elongated shape following the cells’ long axis^{8,9}.”

We also added a Raman spectra of the walls surrounding the ICI and of the ICI previously shown in ex-Supplementary figure 18. This figure is now referenced as Supplementary Fig. 3.

However, abelsonite, mineral deposits of Ni-porphyrin, are known to form in vugs and fractures as a secondary mineral, as a result of migration (e.g., Mason et al., 1989), which would seem to provide a secondary explanation for these inclusions. Additionally, the analyses of the ICI are complicated by the fact that there are not analyses of the matrix. The manuscript notes that Ni-porphyrin complexes are accumulated in sedimentary environments so it is especially important to establish that these ICI features originate from the fossilized organism, and not from diagenetic process. Being able to compare an inside-the-fossil signal with an outside-the-fossil signal would help, but the paper only describes isolated microfossils.

We respectfully disagree. A late contamination by fluids or bitumen would not show similar shapes and distribution within the fossil cells than the reported ICIs. As Rasmussen and collaborators (2021) have shown that in chert, some microfossils can be filled by bitumen and, as such, we must be careful when studying microfossils -or possible microfossils- preserved in chert. If such kind of contamination occurred in our samples preserved as compressions in shales, it would be found in all coexisting microfossils, whatever the species. This is clearly not observed here (Baludikay et al., 2016). It would also be present on the walls, inside and outside of the fossil cells, randomly distributed, with irregular shapes and sizes and not concentrated in the axial plane of the cells. The Mbuji-Mayi Supergroup has been intensively studied by our group. B.K. Baludikay described in his PhD thesis and relevant publications the microfossil assemblages preserved in the several drillcores from the Mbuji-Mayi Supergroup (Baludikay et al., 2016; Baludikay, 2018). He showed that, in the shale

samples from which we extracted the investigated *A. tetragonala* (KN23-123; Baludikay et al., 2018, Baludikay, 2018), solid bitumen occurrences are rare, never associated to the microfossils, and only disseminated in pores and fissures and never associated to the microfossils. Finally, he demonstrated that the acid-maceration do not modify the organic signal in extracted *A. tetragonala* compared to *A. tetragonala* embedded in the rock matrix. Other studies have shown similar results such as Vandenbroucke & Largeau, 2006.

We agree with the reviewer that studying the mineral matrix in which the microfossils are embedded allows distinguishing the processes at the origin of the biogeochemical signatures. In that purpose, our group has carefully studied the chemistry and mineralogy of the embedding matrix (Baludikay et al., 2018, Baludikay, 2018, Baludikay et al., in revision). In parallel to the study of our microfossils, we performed some μ XRF mapping of the matrix when implementing the protocol of analysis. In the surrounding matrix, we do not observe any specific enrichment in nickel and the signal of the fossil are completely diluted in the matrix signal. By extracting the investigated microfossils, we are able to better identify the specimens of *A. tetragonala* and, most importantly, to select the better-preserved specimens containing ICIs. In order to have a control on potential abiotic control on the distribution of metals within the fossil cells, we also investigated specimens without ICIs. We observe that these specimens display no (Suppl. Fig 16), or very low, nickel concentration in the walls (e.g. Suppl. Fig 8). By essence, the transformation of chlorophyll or heme molecules in Ni-porphyrin and/or V-porphyrin is a diagenetic process. In Phanerozoic sediments, porphyrins accumulate in bitumen or crude oils and more rarely in kerogen. The rare bitumen preserved in the shales of the Mbuji-Mayi is overmature, precluding the detection of porphyrins. Moreover, this approach would not link the fossil and the possibly recovered biomarkers, which is the main objective of this study.

The identification of Arctacellularia as algae is also problematic, as it seems to hinge on the morphology of the samples and calculations of estimated Chl content from SR-XRF mapping. The manuscript indicates that these microfossils were previously identified as cyanobacterial or fungal (line 79-80), then describes the morphology of these microfossils as having branching patterns are previously unknown for this taxon (line 63-64). However, it also concludes that these fossils cannot be fungal as branching has not been reported for that particular microfossil (line 87-88). This seems contradictory, especially as later the manuscript notes that fungus sequester Ni as part of their metabolic process (106-107).

A. tetragonala was not previously identified as a *Gloeotrichia spp* or as a spores of fungi, but tentatively interpreted as such. The proposed affinities of *A. tetragonala* by Hofmann & Jackson (1994) and Hermann & Podkovyrov (2008) were solely based on the known morphology of the fossil in microfossil assemblages from the Bylot Supergroup and the Miroedikha Formation, respectively. We clearly state why the combination of characters observed for *A. tetragonala* in the Mbuji-Mayi Supergroup exclude a cyanobacterial affinity. Based on morphology only, we cannot exclude a potential other fungal affinity for *A. tetragonala*. Then, we state that the particular fungal genus to which it could be associated with, as proposed by Hermann & Podkovyrov

(2008), never displays branching. To avoid potential circular reasonings, we needed to find a new independent criterium. By evidencing the presence of tetrapyrrole and by showing that the tetrapyrrole concentration is consistent with chlorophyll, we are able to rule out a fungal affinity for *A. tetragonala*. This is demonstrated by the distribution of Ni in ICI within the fossil cells (SR-XRF) AND the coordination of Ni characteristic of its binding within prophyryns (SR-XANES).

As also suggested by the second reviewer, we added a further discussion in the Extended Data on the affinity of *A. tetragonala* to Archeplastida.

Our reinvestigation of A. tetragonala microfossils evidences that A. tetragonala was a siphonocladous filamentous organism dichotomously branched with equal-diameter branches. The node cell is generally larger than the other cells with a roughly trapezoidal form with two small protuberances to which the branches are attached (Fig. 1). In some specimens, the three branches are still attached (Fig. 1c, 1f), suggesting the nodal cell is not a holdfast structure. A. tetragonala corresponds thus to fragments of a larger organism which could have been either a simple-branched organism or a heterotrichous organism (with a prostrate section and upright sections). However, evidencing a benthic or pelagic habit is not possible in absence of attachment structure (holdfast) or preservation in place in, or on, the substrate. The presence of tetrapyrrole moieties deriving from chlorophyll within A. tetragonala ICIs clearly demonstrates that it was capable of phototrophy and could not be a fungus (obligatory heterotroph) as previously proposed³⁰. Photosynthesis is widespread among eukaryotes³¹ and is found within the supergroups Excavata (Euglenozoans), TSAR (Alveolata and Stramenopiles) and Archaeplastida (Rhodophyta, Chlorophyta and Glaucophyta). Unicellular algae within the eukaryotic supergroups can be ruled out based on the multicellularity of A. tetragonala. Among multicellular filamentous branching algae, Xanthophytes and Phaeophyceans (Stramenopiles) can also be excluded as A. tetragonala specimens do not display dendroid branching, tissue-grade organization, apical septa suggesting apical growth, or typical reproductive structures^{31,32}. Therefore, we assign A. tetragonala to the total group Archaeplastida, where several clades of modern green algae and some florideophyte red algae are known to display a siphonocladous body plan. No distinctive characters of these algae can be found with certainty in A. tetragonala. As such, it is possible that A. tetragonala represents an extinct stem lineage within Archaeplastida and further taxonomic recognition is puzzling. However, the absence of pit plugs, although difficult to observe in fossils, of multiseriate sections in filaments, of longitudinal division, and the fact that the siphonocladous body plan is a derived character in red algae seems to preclude a rhodophycean affinity³³.

Later in the manuscript, the concentration of Ni is used to estimate how much heme and chlorophyll these organisms would have contained, indicating that this concentration suggests that this microfossil is algal. However, this calculation relies on several

assumptions, many unstated, including that each Ni molecule can be directly correlated with a tetrapyrrolic molecule.

The starting hypothesis used to do our calculations of tetrapyrrole content is clearly stated in the supplementary material. The aim of this calculation is to obtain a minimum concentration of the tetrapyrrole molecules within a living cell of *Arctacellularia tetragonala* considering the flattening and the change in volume and size during compaction following the study of Schopf (1992). We assume that all the Ni preserved within the ICIs is linked to the tetrapyrrole moieties based on the XANES spectral shape. This spectral shape is consistent with Ni-porphyrinic species and exclude any other combination of Ni (inorganic oxides, sulphides or organic ligands). In addition, most of the Ni preserved in the ICIs is not endogenic. The inclusion of Ni in the dead cells occurred at the same time than the transformation of biological tetrapyrrole molecules into geoporphyrins, resulting in its fixation within the molecules. The high potential of preservation of Ni-porphyrin and the preserved microenvironment of the fossil have allowed their preservation for more than 1 Ga despite their recombination with the kerogen.

There are other examples of unclear or incomplete data analysis. For instance, when the XANES data are described, the relevant figures show iron sulfide complexes on and in the microfossils (i.e., Fig 2b has sulfide labeled, and the supplementary figures show the co-occurrence of iron and sulfur) but the manuscript describes diagenesis occurring in a non-sulphidic condition (121).

It is important not to confuse global redox conditions and local redox conditions. Sulfides in KN 22 and KN 23 samples are rare and, when present, they are associated to microbial filaments (remnants of microbial mats, Baludikay, 2018). The fact that the Ni is principally linked to organic material implies, in itself, that the redox conditions were non-sulfidic at the moment of its incorporation into the organic material (Algeo & Maynard, 2004). Local sulphidic conditions can happen in the late phase of the diagenesis in pores or during metamorphism when organic sulfide is released from the organic material (Sforna et al., 2014; Sforna et al., 2017). In addition, the sulfides in fig.2 are deposited on the surface of the fossils and it cannot be excluded that they were deposited there during the acid-maceration treatment, storage, and manipulation during sample preparation (as it is the case for the fluorides). We added that the sulphides were attached to the surface of the fossils in the figure caption.

Similarly, although the Raman spectrum is described as showing no evidence for inorganic Ni compounds inside an ICI (Fig S18, line 113-114), it also shows no indication of any Ni-pyrrole compounds. The spectrum shown is labeled as containing D and G carbon bands, there are no bands indicative of a Ni-porphyrin, complexes that have been studied thoroughly by Raman spectroscopy (i.e., Schindler et al., 2018, Ruff data base of Raman spectra of minerals). If these ICI are composed of Ni-porphyrin the fact that Raman spectroscopy does not contain evidence of this composition should be discussed.

We respectfully disagree. The cited paper (Schindler et al., 2018), and most generally any Raman study of porphyrins (e.g. Cantù et al., 2000), are conducted on synthetic/purified porphyrins. Characteristic Raman bands of porphyrins are located between 1350 and 1600 cm^{-1} , which is also the region of vibration of sp^2 carbons in all disordered carbon (kerogen, asphaltene, graphite, e.g. Ferrari & Robertson, 2000; Foucher, 2019). The typical Raman spectrum of kerogen exhibits two main bands generally labelled D, for disordered, and G, for graphite, located respectively around $\sim 1350 \text{ cm}^{-1}$ and $\sim 1600 \text{ cm}^{-1}$ (e.g. Beyssac et al. 2002, 2003; Ferrari 2007; Foucher et al. 2015; Jehlička and Bény 1999; Jehlička et al. 2003; Lahfid et al. 2010; Quirico et al. 2009; Sforza et al. 2014; Foucher, 2019). Sp^2 carbon are very strong Raman scatterers that will be resonant, whatever the excitation laser wavelength (Ferrari, 2007; Foucher, 2019). They will dominate the Raman signal, especially if the other phases vibrating in this region are mineral (mostly lower Raman scatterers) and in trace amounts compared to the kerogen (e.g. Tuschel, 2013). In addition, we don't state that the fossil or the ICIs are made of porphyrins but that the ICIs contain bound porphyrin moieties inside a kerogenous matrix. Therefore, the Raman spectra correspond to Raman spectra of kerogen and not of porphyrins.

Finally, with regards to data analysis and interpretation, the XANES data are described as being "typical of Ni in coordination (IV) in bound N-porphyrinic species" (Line 325) but no line fitting is ever done, instead there is just a stack of spectra to eyeball.

Recording XANES spectra of the ICIs with a very high spatial resolution ($\sim 1\mu\text{m}$) with very low local concentrations is highly demanding and challenging. Indeed, compared to bulk measurements of high concentration samples, there is obviously a higher noise level in the obtained spectra. As such, obtaining micro-XANES spectra is rather unique. Fitting of XANES spectra for very low concentration of the element of interest (13 to 23 ng.cm^{-2}) is also challenging.

We think that the similarity between the spectra of free Ni-porphyrins and asphaltens with our XANES spectra is quite obvious when compared to literature on the different Ni-species. we performed the linear combination fitting requested by the reviewer on our data. We used the standards of inorganic Ni we measured during the experiment and the NiTPP, NiOEP and Asphaltene standards from literature (Nesbitt et al., 2017, Litke, 1986). We did this LCF using the Athena software with the parameters described in Nesbitt et al. (2017). The LCF obtained is mostly good in the edge region and shows a strong influence of the asphaltene (bound tetrapyrroles) in the XANES spectra. There are nonetheless some differences as the incorporation of porphyrins in the kerogen will lead to subtle variations in the bounded metal site chemistry contributing to the observed fluctuations in the sample Ni K-edge XANES spectra. Phenomenon also observed when porphyrins are incorporated in cokes and asphaltens (Nesbitt et al., 2017). In addition, the use of NiTPP and NiOEP as standards to do this LCF could also induce some errors in the fitting as they are model porphyrins and then are different from natural porphyrins. Molecular differences between natural and model porphyrins

can then again add some change in the spectra and then explain the difference between the data and the fitting. Despite these molecular heterogeneities of present Ni-porphyrinic species that would settle the differences in the XANES spectra, our analyses still allow to robustly identify the class of chemical compounds present in our sample, i.e. bounded tetrapyrrole moieties in the kerogen.

We added the LCF in the figures 3 and S18, and added a supplementary table with the results of the LCF to follow reviewer recommendation.

Although this combination of methodologies is intriguing, it is not yet convincing that this is a robust new path forward. In addition to the concerns listed above, the manuscript never describes what other metabolic compounds could be identified with this approach.

We believe that it is out of the scope of this paper. The aim of this paper is to show that it is possible to detect biomarkers within individual fossil cells, making the link between the bulk biomarker record and the fossil record. This approach allows to directly associate a single microfossil and its metabolism, and provides a new tool to elucidate the affinities of enigmatic Proterozoic microfossils. Such approach was never reported before and certainly not in overmature samples. By essence, this innovative approach opens a new and wide window on the diversity of early metabolisms, on the paleobiology and identification of microfossils, and can be applied through the geological record for a better understanding of life evolution.

Reviewer #2 (Remarks to the Author):

Sforna et al. demonstrate in this short communication that intracellular inclusions in the fossil *A. tetragonala* contain Ni-porphyrins likely originating from chlorophyll, thus providing compelling and direct evidence that this organism was a photosynthetic eukaryote.

I consider this an elegant and quite interesting piece of work that, among other things, adds substantial weight to the notion that crown group eukaryotes were already quite diverse by 1.0 Ga; in particular, given that recently there has been somewhat of a case made for a very late radiation of the group. The authors tease the possibility of this approach being applied to much older rocks of Archean age, which I do find quite exciting.

I find the data SR-XRF and XANES data particularly convincing, although I am not myself a practitioner of these methods. Materials and methods are well described in the supplements. I have no major issues with the work.

We thank the reviewer for this favorable review of our manuscript.

Just some minor comments:

Line 39, 164. Wouldn't it be better to exchange here phototrophy; for photosynthetic eukaryotes? It is just that phototrophy is quite a broad term and in this particular case you can be much more specific.

We accordingly changed the manuscript.

In your first and second paragraph, it is sometimes not clear what data is coming from the current work and what data you are citing from previous work as part of introduction and providing context. Could you please just make sure that you delineate and separate better the background work leading to this paper from the new data being reported?

We accordingly changed the manuscript.

Lines 60 to 61. It is unclear whether this interpretation (e.g. multinucleate) is based on refs. 8 and 9 cited in the sentence before, or something that you deduce from the current data. Please reference this better or elaborate on the evidence.

We added a reference (Butterfield et al., 2004).

In any case, are you implying that the ICI is therefore residue from the nucleus? I bring this up because you then go to demonstrate that ICI contains a rather uniform distribution of Ni-tetrapyrroles. In addition, you make a case for the concentrations of these chlorophyll by-products to have decreased by orders of magnitude during diagenesis. Does not this indicate that ICI is more likely to be originating from chloroplast material? Or at least, in this particular case, to also contain substantial chloroplast remnants? Perhaps, you should state this explicitly in the text and what this would imply for other eukaryotic fossils that contain ICI. I am thinking, in particular, of those 1.6 Ga fossils by Bengtson et al. (Plos Biology 2017), who I believe make a case for pyrenoid remnants, which are structures associated with chloroplasts.

Indeed, we think that the tetrapyrrole moieties found in *A. tetragonala* principally derive from the chloroplasts. We cannot completely exclude other heme molecules coming from other organelles but the estimated minimum concentration supports a chlorophyll origin, thus a chloroplast origin. We are not suggesting that these ICI's are nuclei remnants. The actual consensus on the origin of these ICIs is that they are the remnant of the condensed cellular material (cytoplasm + organelles + nucleus) formed before the post-mortem collapse and the diagenetic compaction of the vesicle walls (Pang et al., 2013). Although recent taphonomic experiments (Carlisle et al, 2021) suggest nuclei can be preserved, the peculiar morphology of the long ICI positioned following the main axis the cells in our material precludes a nuclear origin and points towards a cytoplasmic origin. To clarify this, we accordingly modified the text.

“A. tetragonala specimens from the Mbuji-Mayi Supergroup also display a high number of intracellular inclusions (ICI) (Fig.1b, d, h, i). These structures are organic and show the same thermal maturity than the surrounding walls (Supplementary Fig. 3), with greater thickness than the surrounding material. Their shapes and size reflect the ones of the surrounding cells. These characteristics indicate that these ICIs, although taphonomically modified, existed before the diagenetic compaction of the fossils⁸. These ICIs are interpreted as condensed cellular content remains (cytoplasm, organelles and nuclei), because, unlike fossil nuclei, they display an elongated shape following the cells’ long axis^{8,9}.”

Finally, I think the more broader audience interested in the subject would benefit from having access to a summary of the evidence placing *A. tetragonala* within Archeplastida, mentioning whether this is considered to be from a lineage with no easily identifiable living representatives or if it has been placed within a known group. It would also be useful if you can highlight whether there is any conflicting interpretations of fossils identified as *A. tetragonala*. Just adding a paragraph into the Supplementary Information on this, perhaps associated with your Extended Data Table 1, would be nice.

We thank the reviewer for their suggestion and we added a supplementary paragraph in the Supplementary Material (L175-201).

Before this study, only two affinities were proposed for these fossils. Hofmann and Jackson (1994) proposed a cyanobacterial affinity based on the presence of ellipsoidal cells along the uniseriate filaments, comparable to what is observed in the modern cyanobacteria *Gloeotrichia anabaenopsis*. However, based on the combination of characters, we were able to preclude a cyanobacterial affinity (L86-89). The other suggested affinity was given by Hermann and Podkovyrov (2004). They proposed that the cells of *A. tetragonala* were spores of a possible ascomycete by comparing their morphology to the Cretaceous to Pleistocene fossil *Fractisporonites*. By evidencing the presence of tetrapyrrole, and by showing that the tetrapyrrole concentration is consistent with chlorophyll, we are able to rule out a fungal affinity for *A. tetragonala*.

We added a paragraph to the supplementary text discussing more in detail the affinity of *A. tetragonala*, especially why we interpret it as a member of the Archeplastida supergroup. Unfortunately, we did not evidence additional distinctive characters that would allow a further placement in a particular crown lineage within the Archeplastida.

Archeplastida affinity of A. tetragonala

Our reinvestigation of A. tetragonala microfossils evidences that A. tetragonala was a siphonocladous filamentous organism dichotomously branched with equal-diameter branches. The node cell is generally larger than the other cells with a roughly trapezoidal form with two small protuberances to which the branches are attached (Fig. 1). In some specimens, the three branches are still attached (Fig. 1c, 1f), suggesting the nodal cell is not a holdfast structure. A. tetragonala corresponds thus to fragments of a

larger organism which could have been either a simple-branched organism or a heterotrichous organism (with a prostrate section and upright sections). However, evidencing a benthic or pelagic habit is not possible in absence of attachment structure (holdfast) or preservation in place in, or on, the substrate. The presence of tetrapyrrole moieties deriving from chlorophyll within *A. tetragonala* ICIs clearly demonstrates that it was capable of phototrophy and could not be a fungus (obligatory heterotroph) as previously proposed (Hofman & Podkovirov). Photosynthesis is widespread among eukaryotes³¹ and is found within the supergroups Excavata (Euglenozoans), TSAR (Alveolata and Stramenopiles) and Archaeplastida (Rhodophyta, Chlorophyta and Glaucophyta). Unicellular algae within the eukaryotic supergroups can be ruled out based on the multicellularity of *A. tetragonala*. Among multicellular filamentous branching algae, Xanthophytes and Phaeophyceans (Stramenopiles) can also be excluded as *A. tetragonala* specimens do not display dendroid branching, tissue-grade organization, apical septa suggesting apical growth, or typical reproductive structures^{31,32}. Therefore, we assign *A. tetragonala* to the total group Archaeplastida, where several clades of modern green algae and some florideophyte red algae are known to display a siphonocladous body plan. No distinctive characters of these algae can be found with certainty in *A. tetragonala*. As such, it is possible that *A. tetragonala* represents an extinct stem lineage within Archaeplastida and further taxonomic recognition is puzzling. However, the absence of pit plugs, although difficult to observe in fossils, of multiseriate sections in filaments, of longitudinal division, and the fact that the siphonocladous body plan is a derived character in red algae seems to preclude a rhodophycean affinity³³.

Reviewer #1 (Remarks to the Author):

To try to keep this clear, I responded in the original response document. My new comments are in red, and I attached the document below.

Answer to reviewers

Reviewer #1 (Remarks to the Author):

This manuscript describes the application of microscopy, SR-uXRF, and Raman spectroscopy to a total of 56 microfossils. These techniques reveal the presence of inclusions inside microfossils that appear to be composed of Ni-porphyrins. The manuscript describes the morphology of the microfossils as suggestive of algae, the XANES data as indicating the presence of Ni-tetrapyrroles, indicating chlorophyll, and thus argues that the inclusions indicate that these are microfossils of algae that contained chlorophyll. This method is also proposed as being useful for identifying metabolic products in other fossils.

I have several concerns about the manuscript in its present form:

First, the paper never demonstrates that the ICI are syngenetic with the microfossils, rather than things introduced later by diagenetic processes. These inclusions are described as condensed cytoplasm never explained how that is a known fact.

The ICIs from the 65 fossils we studied for this article all answer to the criteria given by Pang et al (2013) to insure that the ICIs are syngenetic and existed before the post-mortem collapse or diagenetic compaction of the vesicle walls. Pang et al. (2013) showed that although ICIs are taphonomically modified (fusion with the cell walls), ICIs existed before the post-mortem collapse or diagenetic compaction of the vesicle walls if the ICIs meet following conditions:

- 1) the size and the shapes of the ICIs reflect the appearance of the surrounding cell
- 2) The ICIs and the enclosing cells show the same answer to any stress field during postmortem of diagenetic deformation
- 3) The ICIs have a greater thickness than the surrounding material and marked by a considerable, granular texture, whereas the rest of the cell appears smoother and flatter.

A large bibliography on Intracellular inclusions (ICIs) does exist and similar inclusions have been documented for many Proterozoic and younger organic-walled microfossils over more than 50 years of palynological studies. Despite an intense debate occurred since their first description in Proterozoic microfossils during the sixties, a consensus on their biologically-derived nature has been reached for organic-walled microfossils in shales (see Pang et al., 2013). ICIs are well-known features occurring in Proterozoic and Phanerozoic organic-walled microfossils both in cherts and shales (e.g. Barghoorn and Schopf, 1965; Pang et al., 2013; Loran and Moczydlowska, 2017; Adam et al., 2017; Baludikay et al., 2016; Beghin et al., 2017; Leiming et al., 2005; Tang et al., 2013; Agic et al., 2017; Grey, 2005; Grey et al., 2003; Prasad et al., 2005; Bengston et al., 2017; Marshall et al., 2017; Wellman et al., 2019; Nowak et al., 2017; etc.). Different origins for these intracellular inclusions have been proposed including diagenetic carbonaceous concretions, taphonomically or biologically condensed cytoplasm, nuclei, mitochondria,

chloroplast, pyrenoids...(e.g. Oehler, 1977, Niklas and Brown, 1981, Dejax et al., 2001, Pang et al., 2013; Knoll & Barghoorn, 1975; Li et al., 2012; Tang et al., 2013; Bengston et al., 2017; Nowak et al., 2017; Moczydlowska, 2016; Carlisle et al, 2021).

To clarify the affinity and syngenicity of these ICIs we accordingly modified the manuscript.

“A. tetragonala specimens from the Mbuji-Mayi Supergroup also display a high number of intracellular inclusions (ICI) (Fig. 1b, d, h, i). These structures are organic and show the same thermal maturity than the surrounding walls (Supplementary Fig. 3), with greater thickness than the surrounding material. Their shapes and size reflect the ones of the surrounding cells. These characteristics indicate that these ICIs, although taphonomically modified, existed before the diagenetic compaction of the fossils⁸. These ICIs are interpreted as condensed cellular content remains (cytoplasm, organelles and nuclei), because, unlike fossil nuclei, they display an elongated shape following the cells’ long axis^{8,9}.”

We also added a Raman spectra of the walls surrounding the ICI and of the ICI previously shown in ex-Supplementary figure 18. This figure is now referenced as Supplementary Fig. 3.

I appreciate the clarification that has been added to the manuscript about the ICIs. However this statement *“These ICIs are interpreted as condensed cellular content remains (cytoplasm, organelles and nuclei), because, unlike fossil nuclei, they display an elongated shape following the cells’ long axis^{8,9}.”*, while it does address an element listed by Pang et al., 2013, does not seem accurate for all fossils depicted in the paper. Supplemental figures 9A, 14A, 14E, 18E, and Figures 1b, 1h all appear to show ICI not aligned with the long axis of the cells. If this is a result of the palynological preparation, that needs to be explained and clarified.

It is true that the Supplementary Figure 3 shows that the Raman spectra of the walls and the inclusion are similar, but that ultimately just reveals that they experienced simultaneous thermal maturation. The thermal history of this unit is referred to as “highly disturbed” in Baludikay et al., 2016, which indicates it is hard to know how long after deposition such thermal alteration could have occurred. Additionally, these spectra look very similar to those collected by Delpomdor et al., 2018 from carbonaceous material found in a fracture in what seems to be the same core.

However, abelsonite, mineral deposits of Ni-porphyrin, are known to form in vugs and fractures as a secondary mineral, as a result of migration (e.g., Mason et al., 1989), which would seem to provide a secondary explanation for these inclusions. Additionally, the analyses of the ICI are complicated by the fact that there are not analyses of the matrix. The manuscript notes that Ni-porphyrin complexes are accumulated in sedimentary environments so it is especially important to establish that these ICI features originate from the fossilized organism, and not from diagenetic process. Being able to compare an inside-the-fossil signal with an outside-the-fossil signal would help, but the paper only describes isolated microfossils.

We respectfully disagree. A late contamination by fluids or bitumen would not show similar shapes and distribution within the fossil cells than the reported ICIs. As Rasmussen and collaborators (2021) have shown that in chert, some microfossils can be filled by bitumen and, as such, we must be careful when studying microfossils -or possible microfossils- preserved in chert. If such kind of contamination occurred in our samples preserved as compressions in shales, it would be found in all coexisting microfossils, whatever the species. This is clearly not observed here (Baludikay et al, 2016).

I would respectfully disagree with the conclusion that this is not observed here. Although Baludikay et al., 2016 only explicitly identifies ICI in specimens of *Arctacellularia tetragonala* (Figurea 8P,R; 9H-J in that paper), many other coexisting species of microfossils pictured in that paper seem to contain visible inclusions (Figures 6K; 7H,I,L-N; 8D; 9M,P; 12L,M). If these are not ICI, I think it is important to further clarify and explain, as it is important to first establish that the features being analyzed are syngenetic.

It would also be present on the walls, inside and outside of the fossil cells, randomly distributed, with irregular shapes and sizes and not concentrated in the axial plane of the cells. The Mbuji-Mayi Supergroup has been intensively studied by our group. B.K. Baludikay described in his PhD thesis and relevant publications the microfossil assemblages preserved in the several drillcores from the Mbuji-Mayi Supergroup (Baludikay et al., 2016; Baludikay, 2018). He showed that, in the shale samples from which we extracted the investigated *A. tetragonala* (KN23-123; Baludikay et al., 2018, Baludikay, 2018), solid bitumen occurrences are rare, never associated to the microfossils, and only disseminated in pores and fissures and never associated to the microfossils. Finally, he demonstrated that the acid-maceration do not modify the organic signal in extracted *A. tetragonala* compared to *A. tetragonala* embedded in the rock matrix. Other studies have shown similar results such as Vandenbroucke & Largeau, 2006.

We agree with the reviewer that studying the mineral matrix in which the microfossils are embedded allows distinguishing the processes at the origin of the biogeochemical signatures. In that purpose, our group has carefully studied the chemistry and mineralogy of the embedding matrix (Baludikay et al., 2018, Baludikay, 2018, Baludikay et al., in revision). In parallel to the study of our microfossils, we performed some μ XRF mapping of the matrix when implementing the protocol of analysis. In the surrounding matrix, we

do not observe any specific enrichment in nickel and the signal of the fossil are completely diluted in the matrix signal.

This data would be a useful addition to the supplementary information as well, as a lack of Ni enrichment in the matrix is a valuable data point.

By extracting the investigated microfossils, we are able to better identify the specimens of *A. tetragonala* and, most importantly, to select the better-preserved specimens containing ICIs. In order to have a control on potential abiotic control on the distribution of metals within the fossil cells, we also investigated specimens without ICIs. We observe that these specimens display no (Suppl. Fig 16), or very low, nickel concentration in the walls (e.g. Suppl. Fig 8). By essence, the transformation of chlorophyll or heme molecules in Ni-porphyrin and/or V-porphyrin is a diagenetic process. In Phanerozoic sediments, porphyrins accumulate in bitumen or crude oils and more rarely in kerogen. The rare bitumen preserved in the shales of the Mbuji-Mayi is overmature, precluding the detection of porphyrins. Moreover, this approach would not link the fossil and the possibly recovered biomarkers, which is the main objective of this study.

As stated above, the Raman spectra collected of the ICI and the walls of the microfossils look very similar to that published on the organic matter found in fractures, and the publication (Delpomdor et al., 2018) indicates that the thermal maturity of this fracture bound organic matter is consistent with that found by Baludikay et al., 2018 (p. 20), so I believe it is important to establish the syngeneity of the material being analyzed, to ensure that the fossil and the poryphyrin should be linked.

The identification of Arctacellularia as algae is also problematic, as it seems to hinge on the morphology of the samples and calculations of estimated Chl content from SR-XRF mapping. The manuscript indicates that these microfossils were previously identified as cyanobacterial or fungal (line 79-80), then describes the morphology of these microfossils as having branching patterns are previously unknown for this taxon (line 63-64). However, it also concludes that these fossils cannot be fungal as branching has not been reported for that particular microfossil (line 87-88). This seems contradictory, especially as later the manuscript notes that fungus sequester Ni as part of their metabolic process (106-107).

A. tetragonala was not previously identified as a *Gloeotrichia spp* or as a spores of fungi, but tentatively interpreted as such. The proposed affinities of *A. tetragonala* by Hofmann & Jackson (1994) and Hermann & Podkovyrov (2008) were solely based on the known morphology of the fossil in microfossil assemblages from the Bylot Supergroup and the Miroedikha Formation, respectively. We clearly state why the combination of characters observed for *A. tetragonala* in the Mbuji-Mayi Supergroup exclude a cyanobacterial affinity. Based on morphology only, we cannot exclude a potential other fungal affinity for *A. tetragonala*. Then, we state that the particular fungal genus to which it could be associated with, as proposed by Hermann & Podkovyrov (2008), never displays

branching. To avoid potential circular reasonings, we needed to find a new independent criterium. By evidencing the presence of tetrapyrrole and by showing that the tetrapyrrole concentration is consistent with chlorophyll, we are able to rule out a fungal affinity for *A. tetragonala*. This is demonstrated by the distribution of Ni in ICI within the fossil cells (SR-XRF) AND the coordination of Ni characteristic of its binding within prophyryns (SR-XANES).

As also suggested by the second reviewer, we added a further discussion in the Extended Data on the affinity of *A. tetragonala* to Archaeplastida.

Our reinvestigation of A. tetragonala microfossils evidences that A. tetragonala was a siphonocladous filamentous organism dichotomously branched with equal-diameter branches. The node cell is generally larger than the other cells with a roughly trapezoidal form with two small protuberances to which the branches are attached (Fig. 1). In some specimens, the three branches are still attached (Fig. 1c, 1f), suggesting the nodal cell is not a holdfast structure. A. tetragonala corresponds thus to fragments of a larger organism which could have been either a simple-branched organism or a heterotrichous organism (with a prostrate section and upright sections). However, evidencing a benthic or pelagic habit is not possible in absence of attachment structure (holdfast) or preservation in place in, or on, the substrate. The presence of tetrapyrrole moieties deriving from chlorophyll within A. tetragonala ICIs clearly demonstrates that it was capable of phototrophy and could not be a fungus (obligatory heterotroph) as previously proposed³⁰. Photosynthesis is widespread among eukaryotes³¹ and is found within the supergroups Excavata (Euglenozoans), TSAR (Alveolata and Stramenopiles) and Archaeplastida (Rhodophyta, Chlorophyta and Glaucophyta). Unicellular algae within the eukaryotic supergroups can be ruled out based on the multicellularity of A. tetragonala. Among multicellular filamentous branching algae, Xanthophytes and Phaeophyceans (Stramenopiles) can also be excluded as A. tetragonala specimens do not display dendroid branching, tissue-grade organization, apical septa suggesting apical growth, or typical reproductive structures^{31,32}. Therefore, we assign A. tetragonala to the total group Archaeplastida, where several clades of modern green algae and some florideophyte red algae are known to display a siphonocladous body plan. No distinctive characters of these algae can be found with certainty in A. tetragonala. As such, it is possible that A. tetragonala represents an extinct stem lineage within Archaeplastida and further taxonomic recognition is puzzling. However, the absence of pit plugs, although difficult to observe in fossils, of multiseriate sections in filaments, of longitudinal division, and the fact that the siphonocladous body plan is a derived character in red algae seems to preclude a rhodophycean affinity³³.

This new discussion helps clarify the identification of the fossils, thank you.

Later in the manuscript, the concentration of Ni is used to estimate how much heme and chlorophyll these organisms would have contained, indicating that this concentration suggests that this microfossil is algal. However, this calculation relies on several assumptions, many unstated, including that each Ni molecule can be directly correlated with a tetrapyrrolic molecule.

The starting hypothesis used to do our calculations of tetrapyrrole content is clearly stated in the supplementary material. The aim of this calculation is to obtain a minimum concentration of the tetrapyrrole molecules within a living cell of *Arctacellularia tetragonala* considering the flattening and the change in volume and size during compaction following the study of Schopf (1992). We assume that all the Ni preserved within the ICIs is linked to the tetrapyrrole moieties based on the XANES spectral shape. This spectral shape is consistent with Ni-porphyrinic species and exclude any other combination of Ni (inorganic oxides, sulphides or organic ligands). In addition, most of the Ni preserved in the ICIs is not endogenic. The inclusion of Ni in the dead cells occurred at the same time than the transformation of biological tetrapyrrole molecules into geoporphyrins, resulting in its fixation within the molecules. The high potential of preservation of Ni-porphyrin and the preserved microenvironment of the fossil have allowed their preservation for more than 1 Ga despite their recombination with the kerogen.

There are other examples of unclear or incomplete data analysis. For instance, when the XANES data are described, the relevant figures show iron sulfide complexes on and in the microfossils (i.e., Fig 2b has sulfide labeled, and the supplementary figures show the co-occurrence of iron and sulfur) but the manuscript describes diagenesis occurring in a non-sulphidic condition (121).

It is important not to confuse global redox conditions and local redox conditions. Sulfides in KN 22 and KN 23 samples are rare and, when present, they are associated to microbial filaments (remnants of microbial mats, Baludikay, 2018). The fact that the Ni is principally linked to organic material implies, in itself, that the redox conditions were non-sulfidic at the moment of its incorporation into the organic material (Algeo & Maynard, 2004). Local sulphidic conditions can happen in the late phase of the diagenesis in pores or during metamorphism when organic sulfide is released from the organic material (Sforna et al., 2014; Sforna et al., 2017). In addition, the sulfides in fig.2 are deposited on the surface of the fossils and it cannot be excluded that they were deposited there during the acid-maceration treatment, storage, and manipulation during sample preparation (as it is the case for the fluorides). We added that the sulphides were attached to the surface of the fossils in the figure caption.

The addition helps clarify this point.

Similarly, although the Raman spectrum is described as showing no evidence for inorganic Ni compounds inside an ICI (Fig S18, line 113-114), it also shows no indication of any Ni-pyrrole compounds. The spectrum shown is labeled as containing D and G carbon bands, there are no bands indicative of a Ni-porphyrin, complexes that have been studied thoroughly by Raman spectroscopy (i.e., Schindler et al., 2018, Ruff data base of Raman spectra of minerals). If these ICI are composed of Ni-porphyrin the fact that Raman spectroscopy does not contain evidence of this composition should be discussed.

We respectfully disagree. The cited paper (Schindler et al., 2018), and most generally any Raman study of porphyrins (e.g. Cantù et al., 2000), are conducted on synthetic/purified porphyrins. Characteristic Raman bands of porphyrins are located between 1350 and 1600 cm^{-1} , which is also the region of vibration of sp^2 carbons in all disordered carbon (kerogen, asphaltene, graphite, e.g. Ferrari & Robertson, 2000; Foucher, 2019). The typical Raman spectrum of kerogen exhibits two main bands generally labelled D, for disordered, and G, for graphite, located respectively around $\sim 1350 \text{ cm}^{-1}$ and $\sim 1600 \text{ cm}^{-1}$ (e.g. Beyssac et al. 2002, 2003; Ferrari 2007; Foucher et al. 2015; Jehlička and Bény 1999; Jehlička et al. 2003; Lahfid et al. 2010; Quirico et al. 2009; Sforza et al. 2014; Foucher, 2019). Sp^2 carbon are very strong Raman scatterers that will be resonant, whatever the excitation laser wavelength (Ferrari, 2007; Foucher, 2019). They will dominate the Raman signal, especially if the other phases vibrating in this region are mineral (mostly lower Raman scatterers) and in trace amounts compared to the kerogen (e.g. Tuschel, 2013). In addition, we don't state that the fossil or the ICIs are made of porphyrins but that the ICIs contain bound porphyrin moieties inside a kerogenous matrix. Therefore, the Raman spectra correspond to Raman spectra of kerogen and not of porphyrins.

Geoporphyrins are highly sensitive to Raman analyses, and have been well characterized in their geological form. While there are characteristic bands that overlap with the D and G bands, there are also bands that occur in other regions, both above and below 1350-1600 cm^{-1} (i.e., <https://rruff.info/S/R070007>). Although the ICI are kerogenous, it is not impossible that Raman analysis would also reveal the presence of a geoporphyrin.

Finally, with regards to data analysis and interpretation, the XANES data are described as being "typical of Ni in coordination (IV) in bound N-porphyrinic species" (Line 325) but no line fitting is ever done, instead there is just a stack of spectra to eyeball.

Recording XANES spectra of the ICIs with a very high spatial resolution ($\sim 1 \mu\text{m}$) with very low local concentrations is highly demanding and challenging. Indeed, compared to bulk measurements of high concentration samples, there is obviously a higher noise level in the obtained spectra. As such, obtaining micro-XANES spectra is rather unique. Fitting of XANES spectra for very low concentration of the element of interest (13 to 23 ng.cm^{-2}) is also challenging.

We think that the similarity between the spectra of free Ni-porphyrins and asphaltens with our XANES spectra is quite obvious when compared to literature on the different Ni-species. we performed the linear combination fitting requested by the reviewer on our data. We used the standards of inorganic Ni we measured during the experiment and the NiTPP, NiOEP and Asphaltene standards from literature (Nesbitt et al., 2017, Litke, 1986). We did this LCF using the Athena software with the parameters described in Nesbitt et al. (2017). The LCF obtained is mostly good in the edge region and shows a strong influence of the asphaltene (bound tetrapyrroles) in the XANES spectra. There are nonetheless some differences as the incorporation of porphyrins in the kerogen will lead to subtle variations in the bounded metal site chemistry contributing to the observed fluctuations in the sample Ni K-edge XANES spectra. Phenomenon also observed when porphyrins are incorporated in cokes and asphaltens (Nesbitt et al., 2017). In addition, the use of NiTPP and NiOEP as standards to do this LCF could also induce some errors in the fitting as they are model porphyrins and then are different from natural porphyrins. Molecular differences between natural and model porphyrins can then again add some change in the spectra and then explain the difference between the data and the fitting. Despite these molecular heterogeneities of present Ni-porphyrinic species that would settle the differences in the XANES spectra, our analyses still allow to robustly identify the class of chemical compounds present in our sample, i.e. bounded tetrapyrrole moieties in the kerogen.

We added the LCF in the figures 3 and S18, and added a supplementary table with the results of the LCF to follow reviewer recommendation.

Thank you for this addition.

Although this combination of methodologies is intriguing, it is not yet convincing that this is a robust new path forward. In addition to the concerns listed above, the manuscript never describes what other metabolic compounds could be identified with this approach.

We believe that it is out of the scope of this paper. The aim of this paper is to show that it is possible to detect biomarkers within individual fossil cells, making the link between the bulk biomarker record and the fossil record. This approach allows to directly associate a single microfossil and its metabolism, and provides a new tool to elucidate the affinities of enigmatic Proterozoic microfossils. Such approach was never reported before and certainly not in overmature samples. By essence, this innovative approach opens a new and wide window on the diversity of early metabolisms, on the paleobiology and identification of microfossils, and can be applied through the geological record for a better understanding of life evolution.

My first concern is still ensuring that the geoporphyryns measured are in features that are biologically derived. There are many ways to introduce geoporphyryns into a sample, and they are very common in petroleum-influenced samples such as these. Providing further data illustrating how these ICI are unique features only associated with these particular microfossils would be an important step, particularly as previous work done on this core by this group show other potential ICI in other microfossils. Similarly, if more data on non-ICI was collected, as indicated above, it would be useful to provide such data.

There are many examples of Precambrian paleontology where chemical analyses create more questions than they answer, and thus it is crucial to establish that the features analyzed are in fact of biological origin and syngenetic with the formation of the unit. I believe there is still work to be done in this regard.

Reviewer #2 (Remarks to the Author):

All of my comments have been addressed satisfactorily. I have no further feedback. Thank you.

Answer to reviewers

We thank the reviewers for their comments. Please find in green our comments for this round of review.

Reviewer #1 (Remarks to the Author):

This manuscript describes the application of microscopy, SR-uXRF, and Raman spectroscopy to a total of 56 microfossils. These techniques reveal the presence of inclusions inside microfossils that appear to be composed of Ni-porphyrins. The manuscript describes the morphology of the microfossils as suggestive of algae, the XANES data as indicating the presence of Ni-tetrapyrroles, indicating chlorophyll, and thus argues that the inclusions indicate that these are microfossils of algae that contained chlorophyll. This method is also proposed as being useful for identifying metabolic products in other fossils.

I have several concerns about the manuscript in its present form:

First, the paper never demonstrates that the ICI are syngenetic with the microfossils, rather than things introduced later by diagenetic processes. These inclusions are described as condensed cytoplasm never explained how that is a known fact.

The ICIs from the 65 fossils we studied for this article all answer to the criteria given by Pang et al (2013) to insure that the ICIs are syngenetic and existed before the post-mortem collapse or diagenetic compaction of the vesicle walls. Pang et al. (2013) showed that although ICIs are taphonomically modified (fusion with the cell walls), ICIs existed before the post-mortem collapse or diagenetic compaction of the vesicle walls if the ICIs meet following conditions:

- 1) the size and the shapes of the ICIs reflect the appearance of the surrounding cell
- 2) The ICIs and the enclosing cells show the same answer to any stress field during postmortem of diagenetic deformation
- 3) The ICIs have a greater thickness than the surrounding material and marked by a considerable, granular texture, whereas the rest of the cell appears smoother and flatter.

A large bibliography on Intracellular inclusions (ICIs) does exist and similar inclusions have been documented for many Proterozoic and younger organic-walled microfossils over more than 50 years of palynological studies. Despite an intense debate occurred since their first description in Proterozoic microfossils during the sixties, a consensus on their biologically-derived nature has been reached for organic-walled microfossils in shales (see Pang et al., 2013). ICIs are well-known features occurring in Proterozoic and Phanerozoic organic-walled microfossils both in cherts and shales (e.g. Barghoorn and Schopf, 1965; Pang et al., 2013; Loron and Moczydłowska, 2017; Adam et al., 2017;

Baludikay et al., 2016; Beghin et al., 2017; Leiming et al., 2005; Tang et al., 2013; Agic et al., 2017; Grey, 2005; Grey et al., 2003; Prasad et al., 2005; Bengston et al., 2017; Marshall et al., 2017; Wellman et al., 2019; Nowak et al., 2017; etc.). Different origins for these intracellular inclusions have been proposed including diagenetic carbonaceous concretions, taphonomically or biologically condensed cytoplasm, nuclei, mitochondria, chloroplast, pyrenoids...(e.g. Oehler, 1977, Niklas and Brown, 1981, Dejax et al., 2001, Pang et al., 2013; Knoll & Barghoorn, 1975; Li et al., 2012; Tang et al., 2013; Bengston et al., 2017; Nowak et al., 2017; Moczydlowska, 2016; Carlisle et al., 2021).

To clarify the affinity and syngenicity of these ICIs we accordingly modified the manuscript.

“A. tetragonala specimens from the Mbuji-Mayi Supergroup also display a high number of intracellular inclusions (ICI) (Fig. 1b, d, h, i). These structures are organic and show the same thermal maturity than the surrounding walls (Supplementary Fig. 3), with greater thickness than the surrounding material. Their shapes and size reflect the ones of the surrounding cells. These characteristics indicate that these ICIs, although taphonomically modified, existed before the diagenetic compaction of the fossils⁸. These ICIs are interpreted as condensed cellular content remains (cytoplasm, organelles and nuclei), because, unlike fossil nuclei, they display an elongated shape following the cells’ long axis^{8,9}.”

We also added a Raman spectra of the walls surrounding the ICI and of the ICI previously shown in ex-Supplementary figure 18. This figure is now referenced as Supplementary Fig. 3.

I appreciate the clarification that has been added to the manuscript about the ICIs. However this statement *“These ICIs are interpreted as condensed cellular content remains (cytoplasm, organelles and nuclei), because, unlike fossil nuclei, they display an elongated shape following the cells’ long axis^{8,9}.”*, while it does address an element listed by Pang et al., 2013, does not seem accurate for all fossils depicted in the paper. Supplemental figures 9A, 14A, 14E, 18E, and Figures 1b, 1h all appear to show ICI not aligned with the long axis of the cells. If this is a result of the palynological preparation, that needs to be explained and clarified.

As noted by the reviewer, some of the investigated ICIs are not aligned along the cell axis, can be “distorted” or have shapes that do not clearly reflect the shape of the cell. If some specimens can become teared, folded, ruffled, or rolled up during laboratory preparation and mounting, most of these features were most probably acquired with the compression during burial, (Grey and Wilman, 2009). The adapted preparation method used in Early Life lab (Grey, 1999) is a low manipulation protocol without centrifugation, in order to lower the risk of specimen degradations. Fossils preserved as flattened two-dimensional structures exhibit compression-produced folds, pleats and flexures (e.g.,

Schopf, 1992 and all shale-hosted microfossils assemblage descriptions for the Proterozoic and Phanerozoic). As a consequence, the shape and the position of these ICIs could have been modified during the compression rather than during the palynological preparation. These “deformed” or “misplaced” ICIs can still be related to the other ICIs if they still respect the deformation and compression planes (Pang et al., 2013) as it is the case here.

It is true that the Supplementary Figure 3 shows that the Raman spectra of the walls and the inclusion are similar, but that ultimately just reveals that they experienced simultaneous thermal maturation. The thermal history of this unit is referred to as “highly disturbed” in Baludikay et al., 2016, which indicates it is hard to know how long after deposition such thermal alteration could have occurred.

Additionally, these spectra look very similar to those collected by Delpomdor et al., 2018 from carbonaceous material found in a fracture in what seems to be the same core.

The basin has indeed a complex thermal history linked with the different phases of deformation that it underwent. However, locally we can constrain this history.

Delpomdor and colleagues do not indicate the depth from which the samples they studied come from. We also did not find any correspondence in all other articles written by this group as well as in Delpomdor’s PhD manuscript. Although our samples and theirs are coming from the same formation, BIIc, we cannot directly link them together, or only tentatively. It is nonetheless important to point out that we do not have large fracturing veins in our samples. If fractures occur, they are small and only contain very few carbonaceous material associated to marcasite inside (C. François, pers. Comm.).

To strengthen our position, we nonetheless compared our spectra to the only spectrum of the late asphaltite described in cracks and reported by Delpomdor et al., 2018 and we respectfully disagree with the reviewer on their resemblance. Our spectra differ from theirs as it shows a narrower D1 (~115 cm⁻¹ here vs ~140 cm⁻¹ in Delpomdor et al) and a higher ID1/IG ratio (~2 here vs 0.82 in Delpomdor et al). Such differences are significative of a higher thermal maturity for the fossil kerogen compared to the late contaminating asphaltite (e.g. Lahfid et al., 2010; Sforza et al., 2014; Kouketsu et al., 2014 and reference therein). The Raman reflectance measured from our data has a mean of ~2.29, corresponding to a mean burial peak temperature of ~205°C (see Baludikay et al., 2018 for calculation steps). This is very similar to what was found by Baludikay et al., 2018, both for the fossils and the amorphous organic matter contained in KN23. These values differ from what was previously reported in the Vienna abstract (Baludikay et al., 2016, EGU abstracts) cited by Delpomdor et al., 2018 as reference for the Kanshi drillcore values, because the thermal maturity of the carbonaceous material of the Kanshi drillcore was reinvestigated and the geothermometry was recalculated in the months following the conference for which the abstract was submitted.

Delpomdor and colleagues report only the mean values in their manuscript and not the raw spectra. Therefore, we recalculated the Raman reflectance for the spectrum reported in their 2018 paper. We obtained a Raman reflectance of 0.9, corresponding to a burial peak temperature of ~120°C, well below our values (see above). Based on the Raman investigation (Supplementary Table 2), both the fossils and their ICIs have seen the same thermal maturation history. As a conclusion, we can vigorously exclude that the material contained inside the cells originate from the post-Proterozoic oil migration that has given rise to the asphaltite reported by Delpomdor et al., 2018.

We added in the supplementary material a new table reporting several points of Raman performed on the ICIs and the cell walls and added in the text a new paragraph discussing the syngenicity of the porphyrins (I175-198).

“Origin of the Ni-tetrapyrrole moieties

As Ni-tetrapyrrole are common molecules in Phanerozoic petroleum and can be found up to 1.1 Ga², it is important to assess the syngenicity of the tetrapyrrole recovered in the microfossils. Oil migration from a late Neoproterozoic/Early Palaeozoic active petroleum system has been observed in the Sankuru-Mbuji-Mayi-Lomami-Lovoy⁴⁵ and is thought to have occurred during the Cambrian-Ordovician. As a result of this late oil migration, solid asphaltites have been reported in the BIIc dolomites in cracks and as inclusions. However, our Raman spectra indicate that the ICIs experienced the same thermal maturation than the walls (Supplementary Fig. 4, and Supplementary Table 2). In addition, the only reported Raman spectrum by (45) on these asphaltites shows a less mature signature than the kerogen in KN 22 and KN23⁴⁵. This excludes a contamination of the ICIs during the oil migration episode. Synsedimentary bitumen were also reported in the Mbuji-Mayi shales but are not associated with the microfossils⁶, as observed, for instance, in the Gunflint chert microfossils⁴⁶, where bitumen fills the silicified microfossils preserved in 3D. The context of deposition and preservation in shales (carbonaceous fossils compressed in 2D parallel to the bedding with continuous walls) is indeed less favourable for bitumen migration than in chert where the bitumen migrates through the silicifying microfossils and fills semi-hollow molds⁴⁶. Moreover, the distribution of ICI relative to the fossils is non-random, on the contrary to migrating oil. The presence of a single ICI inside cells (only 1 per cell), their absence on the outer fossil walls, their nearly constant shape, and their occurrence in only a few species in the assemblage (including *A. tetragonala*⁷), and not all specimens of these species exclude a bitumen origin. In *A. tetragonala*, the tetrapyrroles moieties are specifically associated with the ICIs and not with the walls (either from ICI-containing or empty cell walls) as showed by the Ni distribution in the cells. As these ICIs are considered structures forming before or during burial⁸, the Ni-tetrapyrrole moieties are syngenic with these ICIs, originating from the living organisms and not from a late contamination during oil migration.”

However, abelsonite, mineral deposits of Ni-porphyrin, are known to form in vugs and fractures as a secondary mineral, as a result of migration (e.g., Mason et al., 1989), which would seem to provide a secondary explanation for these inclusions. Additionally, the analyses of the ICI are complicated by the fact that there are not analyses of the matrix. The manuscript notes that Ni-porphyrin complexes are accumulated in sedimentary environments so it is especially important to establish that these ICI features originate from the fossilized organism, and not from diagenetic process. Being able to compare an inside-the-fossil signal with an outside-the-fossil signal would help, but the paper only describes isolated microfossils.

We respectfully disagree. A late contamination by fluids or bitumen would not show similar shapes and distribution within the fossil cells than the reported ICIs. As Rasmussen and collaborators (2021) have shown that in chert, some microfossils can be filled by bitumen and, as such, we must be careful when studying microfossils -or possible microfossils- preserved in chert. If such kind of contamination occurred in our samples preserved as compressions in shales, it would be found in all coexisting microfossils, whatever the species. This is clearly not observed here (Baludikay et al, 2016).

I would respectfully disagree with the conclusion that this is not observed here. Although Baludikay et al., 2016 only explicitly identifies ICI in specimens of *Arctacellularia tetragonala* (Figurea 8P,R; 9H-J in that paper), many other coexisting species of microfossils pictured in that paper seem to contain visible inclusions (Figures 6K; 7H,I,L-N; 8D; 9M,P; 12L,M). If these are not ICI,I think it is important to further clarify and explain, as it is important to first establish that the features being analyzed are syngenetic.

The Mbuji-Mayi fossil assemblage has several fossil taxa that contain intracellular inclusions, such as *P. filiformis*, *Chlorogloeaopsis spp*, *J. solubila*, *Synsphaeridium spp* or *Leiospheridia spp*, in addition of *A. tetragonala*. Interestingly, these taxa also often display these ICIs in other fossil assemblages of the same period such as for example the Atar/Al Mreiti Group (Beghin et al. , 2017); the Shaler Supergroup (Loron et al., 2019), the Liulaobei Formation (Pang et al., 2013);the Lakhanda Fm (German, 1990), also in younger Neoproterozoic successions (Porter and Riedman, 2016; Butterfield et al., 1994) and some of these taxa also show ICI in older successions; e.g. Roper Group (Javaux & Knoll, 2017), and other formations (Jankauskas et al., 1989). The presence of these ICIs is then a common feature in these taxa, and their study is the subject of an upcoming article in preparation.

Other older taxa (e.g. *Dictyosphaera delicata* and *Shuiyousphaeridium macroreticulatum*, Pang et al., 2013; Adam et al., 2017) and younger taxa (e.g. *Valkyria borealis*, *Germinosphaera fibrilla*, Butterfield et al., 1994) also display this ICIs. In modern algae,

ICIs-look alike are formed during desiccation as it has been observed for *Zygonema* (Lajos et al., 2016).

We understand the concerns of the reviewer but no process, even in “petrol-influenced” systems, could explain why only certain taxa in a given fossiliferous horizon would be contaminated and why only certain specimens of the given taxa. Based on all those lines of evidence, we conclude that the ICIs in the studied *Arctacellularia* are syngenetic features that formed post-mortem, either before the burial or during the burial, and that were accordingly deformed and compressed.

We accordingly modified the paragraph about the ICIs (l61-77)

A. tetragonala is ubiquitously found in coeval sedimentary deposits from the Late Mesoproterozoic to early Neoproterozoic from Siberia, Canada, West and Central Africa, China, and India^{6,9,7} (for review see Supplementary Table 1). *A. tetragonala* is a readily identifiable microfossil, consisting of unsheathed chains (filaments) of barrel-shaped cells, with deep constrictions between the cells and terminal lanceolate folds at cell ends (Fig. 1, Supplementary Fig. 2). In the Mbuji-Mayi population we studied (n=65, Supplementary Fig. 2), fragments of uniseriate filaments (1 to 25 cells) may reach up to 550 µm in length. Cells within these filaments have a similar width (25-45 µm) but variable length (15-100 µm). *A. tetragonala* specimens from the Mbuji-Mayi Supergroup also display a high number of intracellular inclusions (ICI) (Fig.1b, d, h, i). These structures are organic and show the same thermal maturity than the surrounding walls (Supplementary Fig. 3, Supplementary Table 2). In addition, these intracellular structures have a greater thickness than the surrounding material and their shapes, size and flattening plan reflect the ones of the surrounding cells. These characteristics indicate that these ICIs, although taphonomically modified, are syngenetic with the fossils and existed before the diagenetic compaction of the fossils⁸. ICIs are common features in shale-hosted assemblages throughout the Proterozoic and the Phanerozoic^{7, 9-19}. Despite an intense debate occurred since their first description in Proterozoic microfossils²⁰, a consensus on their biologically derived nature has been reached for organic-walled microfossils in shales^{8,21}.

It would also be present on the walls, inside and outside of the fossil cells, randomly distributed, with irregular shapes and sizes and not concentrated in the axial plane of the cells. The Mbuji-Mayi Supergroup has been intensively studied by our group. B.K. Baludikay described in his PhD thesis and relevant publications the microfossil assemblages preserved in the several drillcores from the Mbuji-Mayi Supergroup (Baludikay et al., 2016; Baludikay, 2018). He showed that, in the shale samples from which we extracted the investigated *A. tetragonala* (KN23-123; Baludikay et al., 2018, Baludikay, 2018), solid bitumen occurrences are rare, never associated to the microfossils, and only disseminated in pores and fissures and never associated to the

microfossils. Finally, he demonstrated that the acid-maceration do not modify the organic signal in extracted *A. tetragonala* compared to *A. tetragonala* embedded in the rock matrix. Other studies have shown similar results such as Vandembroucke & Largeau, 2006.

We agree with the reviewer that studying the mineral matrix in which the microfossils are embedded allows distinguishing the processes at the origin of the biogeochemical signatures. In that purpose, our group has carefully studied the chemistry and mineralogy of the embedding matrix (Baludikay et al., 2018, Baludikay, 2018, Baludikay et al., in revision). In parallel to the study of our microfossils, we performed some μ XRF mapping of the matrix when implementing the protocol of analysis. In the surrounding matrix, we do not observe any specific enrichment in nickel and the signal of the fossil are completely diluted in the matrix signal.

This data would be a useful addition to the supplementary information as well, as a lack of Ni enrichment in the matrix is a valuable data point.

We added a paragraph in the supplementary data (L51-59) as well as two additional supplementary figures.

“Petrography of the mineral matrix

The petrographical study of the mineral matrix of KN22 and KN23 shales shows that the matrix consists mostly of quartz and clays with presence of K-feldspath, calcite, anatase, and rare sulfides (<2%), which are mostly marcasite. Amorphous organic matter is found mostly associated to marcasite. Previous study led in the Early Life laboratory has shown that the clays are a mixture of illite-micas, chlorite, kaolinite and 10-14 mixed layer clays¹⁶. The SR- μ XRF mapping (Supplementary Fig. 20) confirms this petrological analysis, as shown by the distribution of Fe and K. The distribution of Ti highlights the presence of anatase, which also contains some V. The Ca map shows the limited carbonate distribution. Sulfides contain low amounts of Ni, Cu, As and Cr.”

By extracting the investigated microfossils, we are able to better identify the specimens of *A. tetragonala* and, most importantly, to select the better-preserved specimens containing ICIs. In order to have a control on potential abiotic control on the distribution of metals within the fossil cells, we also investigated specimens without ICIs. We observe that these specimens display no (Suppl. Fig 16), or very low, nickel concentration in the walls (e.g. Suppl. Fig 8). By essence, the transformation of chlorophyll or heme molecules in Ni-porphyrin and/or V-porphyrin is a diagenetic process. In Phanerozoic sediments, porphyrins accumulate in bitumen or crude oils and more rarely in kerogen. The rare bitumen preserved in the shales of the Mbuji-Mayi is overmature, precluding the detection

of porphyrins. Moreover, this approach would not link the fossil and the possibly recovered biomarkers, which is the main objective of this study.

As stated above, the Raman spectra collected of the ICI and the walls of the microfossils look very similar to that published on the organic matter found in fractures, and the publication (Delpomdor et al., 2018) indicates that the thermal maturity of this fracture bound organic matter is consistent with that found by Baludikay et al., 2018 (p. 20), so I believe it is important to establish the syngeneity of the material being analyzed, to ensure that the fossil and the porphyrin should be linked.

Please see our answer above.

The identification of Arctacellularia as algae is also problematic, as it seems to hinge on the morphology of the samples and calculations of estimated Chl content from SR-XRF mapping. The manuscript indicates that these microfossils were previously identified as cyanobacterial or fungal (line 79-80), then describes the morphology of these microfossils as having branching patterns are previously unknown for this taxon (line 63-64). However, it also concludes that these fossils cannot be fungal as branching has not been reported for that particular microfossil (line 87-88). This seems contradictory, especially as later the manuscript notes that fungus sequester Ni as part of their metabolic process (106-107).

A. tetragonala was not previously identified as a *Gloeotrichia spp* or as a spores of fungi, but tentatively interpreted as such. The proposed affinities of *A. tetragonala* by Hofmann & Jackson (1994) and Hermann & Podkovyrov (2008) were solely based on the known morphology of the fossil in microfossil assemblages from the Bylot Supergroup and the Miroedikha Formation, respectively. We clearly state why the combination of characters observed for *A. tetragonala* in the Mbuji-Mayi Supergroup exclude a cyanobacterial affinity. Based on morphology only, we cannot exclude a potential other fungal affinity for *A. tetragonala*. Then, we state that the particular fungal genus to which it could be associated with, as proposed by Hermann & Podkovyrov (2008), never displays branching. To avoid potential circular reasonings, we needed to find a new independent criterium. By evidencing the presence of tetrapyrrole and by showing that the tetrapyrrole concentration is consistent with chlorophyll, we are able to rule out a fungal affinity for *A. tetragonala*. This is demonstrated by the distribution of Ni in ICI within the fossil cells (SR-XRF) AND the coordination of Ni characteristic of its binding within porphyrins (SR-XANES).

As also suggested by the second reviewer, we added a further discussion in the Extended Data on the affinity of *A. tetragonala* to Archeplastida.

Our reinvestigation of *A. tetragonala* microfossils evidences that *A. tetragonala* was a siphonocladous filamentous organism dichotomously branched with equal-diameter branches. The node cell is generally larger than the other cells with a roughly trapezoidal form with two small protuberances to which the branches are attached (Fig. 1). In some specimens, the three branches are still attached (Fig. 1c, 1f), suggesting the nodal cell is not a holdfast structure. *A. tetragonala* corresponds thus to fragments of a larger organism which could have been either a simple-branched organism or a heterotrichous organism (with a prostrate section and upright sections). However, evidencing a benthic or pelagic habit is not possible in absence of attachment structure (holdfast) or preservation in place in, or on, the substrate. The presence of tetrapyrrole moieties deriving from chlorophyll within *A. tetragonala* ICIs clearly demonstrates that it was capable of phototrophy and could not be a fungus (obligatory heterotroph) as previously proposed³⁰. Photosynthesis is widespread among eukaryotes³¹ and is found within the supergroups Excavata (Euglenozoans), TSAR (Alveolata and Stramenopiles) and Archaeplastida (Rhodophyta, Chlorophyta and Glaucophyta). Unicellular algae within the eukaryotic supergroups can be ruled out based on the multicellularity of *A. tetragonala*. Among multicellular filamentous branching algae, Xanthophytes and Phaeophyceans (Stramenopiles) can also be excluded as *A. tetragonala* specimens do not display dendroid branching, tissue-grade organization, apical septa suggesting apical growth, or typical reproductive structures^{31,32}. Therefore, we assign *A. tetragonala* to the total group Archaeplastida, where several clades of modern green algae and some florideophyte red algae are known to display a siphonocladous body plan. No distinctive characters of these algae can be found with certainty in *A. tetragonala*. As such, it is possible that *A. tetragonala* represents an extinct stem lineage within Archaeplastida and further taxonomic recognition is puzzling. However, the absence of pit plugs, although difficult to observe in fossils, of multiseriate sections in filaments, of longitudinal division, and the fact that the siphonocladous body plan is a derived character in red algae seems to preclude a rhodophycean affinity³³.

This new discussion helps clarify the identification of the fossils, thank you.

Later in the manuscript, the concentration of Ni is used to estimate how much heme and chlorophyll these organisms would have contained, indicating that this concentration suggests that this microfossil is algal. However, this calculation relies on several assumptions, many unstated, including that each Ni molecule can be directly correlated with a tetrapyrrolic molecule.

The starting hypothesis used to do our calculations of tetrapyrrole content is clearly stated in the supplementary material. The aim of this calculation is to obtain a minimum concentration of the tetrapyrrole molecules within a living cell of *Arctacellularia tetragonala* considering the flattening and the change in volume and size during compaction following the study of Schopf (1992). We assume that all the Ni preserved within the ICIs is linked

to the tetrapyrrole moieties based on the XANES spectral shape. This spectral shape is consistent with Ni-porphyrinic species and exclude any other combination of Ni (inorganic oxides, sulphides or organic ligands). In addition, most of the Ni preserved in the ICIs is not endogenic. The inclusion of Ni in the dead cells occurred at the same time than the transformation of biological tetrapyrrole molecules into geoporphyrins, resulting in its fixation within the molecules. The high potential of preservation of Ni-porphyrin and the preserved microenvironment of the fossil have allowed their preservation for more than 1 Ga despite their recombination with the kerogen.

There are other examples of unclear or incomplete data analysis. For instance, when the XANES data are described, the relevant figures show iron sulfide complexes on and in the microfossils (i.e., Fig 2b has sulfide labeled, and the supplementary figures show the co-occurrence of iron and sulfur) but the manuscript describes diagenesis occurring in a non-sulphidic condition (121).

It is important not to confuse global redox conditions and local redox conditions. Sulfides in KN 22 and KN 23 samples are rare and, when present, they are associated to microbial filaments (remnants of microbial mats, Baludikay, 2018). The fact that the Ni is principally linked to organic material implies, in itself, that the redox conditions were non-sulfidic at the moment of its incorporation into the organic material (Algeo & Maynard, 2004). Local sulphidic conditions can happen in the late phase of the diagenesis in pores or during metamorphism when organic sulfide is released from the organic material (Sforna et al., 2014; Sforna et al., 2017). In addition, the sulfides in fig.2 are deposited on the surface of the fossils and it cannot be excluded that they were deposited there during the acid-maceration treatment, storage, and manipulation during sample preparation (as it is the case for the fluorides). We added that the sulphides were attached to the surface of the fossils in the figure caption.

The addition helps clarify this point.

Similarly, although the Raman spectrum is described as showing no evidence for inorganic Ni compounds inside an ICI (Fig S18, line 113-114), it also shows no indication of any Ni-pyrrole compounds. The spectrum shown is labeled as containing D and G carbon bands, there are no bands indicative of a Ni-porphyrin, complexes that have been studied thoroughly by Raman spectroscopy (i.e., Schindler et al., 2018, Ruff data base of Raman spectra of minerals). If these ICI are composed of Ni-porphyrin the fact that Raman spectroscopy does not contain evidence of this composition should be discussed.

We respectfully disagree. The cited paper (Schindler et al., 2018), and most generally any Raman study of porphyrins (e.g. Cantù et al., 2000), are conducted on synthetic/purified porphyrins. Characteristic Raman bands of porphyrins are located between 1350 and 1600 cm^{-1} , which is also the region of vibration of sp^2 carbons in all disordered carbon (kerogen, asphaltene, graphite, e.g. Ferrari & Robertson, 2000;

Foucher, 2019). The typical Raman spectrum of kerogen exhibits two main bands generally labelled D, for disordered, and G, for graphite, located respectively around $\sim 1350\text{ cm}^{-1}$ and $\sim 1600\text{ cm}^{-1}$ (e.g. Beyssac et al. 2002, 2003; Ferrari 2007; Foucher et al. 2015; Jehlička and Bény 1999; Jehlička et al. 2003; Lahfid et al. 2010; Quirico et al. 2009; Sforza et al. 2014; Foucher, 2019). Sp^2 carbon are very strong Raman scatterers that will be resonant, whatever the excitation laser wavelength (Ferrari, 2007; Foucher, 2019). They will dominate the Raman signal, especially if the other phases vibrating in this region are mineral (mostly lower Raman scatterers) and in trace amounts compared to the kerogen (e.g. Tuschel, 2013). In addition, we don't state that the fossil or the ICIs are made of porphyrins but that the ICIs contain bound porphyrin moieties inside a kerogenous matrix. Therefore, the Raman spectra correspond to Raman spectra of kerogen and not of porphyrins.

Geoporphyrins are highly sensitive to Raman analyses, and have been well characterized in their geological form. While there are characteristic bands that overlap with the D and G bands, there are also bands that occur in other regions, both above and below $1350\text{-}1600\text{ cm}^{-1}$ (i.e., <https://ruff.info/S/R070007>). Although the ICIs are kerogenous, it is not impossible that Raman analysis would also reveal the presence of a geoporphyrin.

We acquired spectra with a 514 nm laser for both porphyrin standards, several points in the walls and in the ICIS, as well as an hyperspectral map on specimens of *A. tetragonala*. Similarly, we acquired them with a 785 nm laser as it seems, from the standard the reviewer gave us, that this laser is more sensible to the porphyrin outside of the kerogen signature region. Again, no signal was recovered. Our Raman analyses did not evidence porphyrin peaks as we expected. As we previously stated the kerogen signal dominates the Raman spectra as the kerogen forming the microfossils is much more abundant than the porphyrins (a few ng/cm^2). Unfortunately, we cannot evidence porphyrins directly in Precambrian microfossils simply using Raman spectroscopy, and the higher resolution technique we implemented in this article is a very potent technique to do it.

Finally, with regards to data analysis and interpretation, the XANES data are described as being "typical of Ni in coordination (IV) in bound N-porphyrinic species" (Line 325) but no line fitting is ever done, instead there is just a stack of spectra to eyeball.

Recording XANES spectra of the ICIs with a very high spatial resolution ($\sim 1\mu\text{m}$) with very low local concentrations is highly demanding and challenging. Indeed, compared to bulk measurements of high concentration samples, there is obviously a higher noise level in the obtained spectra. As such, obtaining micro-XANES spectra is rather unique. Fitting of XANES spectra for very low concentration of the element of interest ($13\text{ to }23\text{ ng.cm}^{-2}$) is also challenging.

We think that the similarity between the spectra of free Ni-porphyrins and asphaltens with our XANES spectra is quite obvious when compared to literature on the different Ni-

species. we performed the linear combination fitting requested by the reviewer on our data. We used the standards of inorganic Ni we measured during the experiment and the NiTPP, NiOEP and Asphaltene standards from literature (Nesbitt et al., 2017, Litke, 1986). We did this LCF using the Athena software with the parameters described in Nesbitt et al. (2017). The LCF obtained is mostly good in the edge region and shows a strong influence of the asphaltene (bound tetrapyrrolles) in the XANES spectra. There are nonetheless some differences as the incorporation of porphyrins in the kerogen will lead to subtle variations in the bounded metal site chemistry contributing to the observed fluctuations in the sample Ni K-edge XANES spectra. Phenomenon also observed when porphyrins are incorporated in cokes and asphaltens (Nesbitt et al., 2017). In addition, the use of NiTPP and NiOEP as standards to do this LCF could also induce some errors in the fitting as they are model porphyrins and then are different from natural porphyrins. Molecular differences between natural and model porphyrins can then again add some change in the spectra and then explain the difference between the data and the fitting. Despite these molecular heterogeneities of present Ni-porphyrinic species that would settle the differences in the XANES spectra, our analyses still allow to robustly identify the class of chemical compounds present in our sample, i.e. bounded tetrapyrrole moieties in the kerogen.

We added the LCF in the figures 3 and S18, and added a supplementary table with the results of the LCF to follow reviewer recommendation.

Thank you for this addition.

Although this combination of methodologies is intriguing, it is not yet convincing that this is a robust new path forward. In addition to the concerns listed above, the manuscript never describes what other metabolic compounds could be identified with this approach.

We believe that it is out of the scope of this paper. The aim of this paper is to show that it is possible to detect biomarkers within individual fossil cells, making the link between the bulk biomarker record and the fossil record. This approach allows to directly associate a single microfossil and its metabolism, and provides a new tool to elucidate the affinities of enigmatic Proterozoic microfossils. Such approach was never reported before and certainly not in overmature samples. By essence, this innovative approach opens a new and wide window on the diversity of early metabolisms, on the paleobiology and identification of microfossils, and can be applied through the geological record for a better understanding of life evolution.

My first concern is still ensuring that the geoporphyrins measured are in features that are biologically derived. There are many ways to introduce geoporphyrins into a sample, and they are very common in petroleum-influenced samples such as these. Providing further data illustrating how these ICI are unique features only associated with these particular microfossils would be an important step, particularly as previous work done on this core

by this group show other potential ICI in other microfossils. Similarly, if more data on non-ICI was collected, as indicated above, it would be useful to provide such data.

There are many examples of Precambrian paleontology where chemical analyses create more questions than they answer, and thus it is crucial to establish that the features analyzed are in fact of biological origin and syngenetic with the formation of the unit. I believe there is still work to be done in this regard.

We have clearly addressed the two main concerns of the reviewer -- the syngeneticity of ICIs and the syngeneticity of the porphyrins -- by adding a new paragraph addressing the syngeneticity of the porphyrins and by completing the paragraph about the ICIs.

REVIEWERS' COMMENTS

Reviewer #1 (Remarks to the Author):

After the latest round of revisions I am satisfied that my original concerns were addressed, either with new information or by better presenting possible alternative conclusions/interpretations.

Thank you for the additional work you did to address these concerns

Answer to the reviewers

We thank both of the reviewers for their constructive comments that helped to strengthen this manuscript.